EMBO
Molecular Medicine

# Targeting *miR-223* in neutrophils enhances the clearance of *Staphylococcus aureus* in infected wounds

Maiko de Kerckhove[1], Katsuya Tanaka[1,2,3], Takahiro Umehara[4], Momoko Okamoto[1,5], Sotaro Kanematsu[6], Hiroko Hayashi[1], Hiroki Yano[2], Soushi Nishiura[1], Shiho Tooyama[1], Yutaka Matsubayashi[7,8], Toshimitsu Komatsu[1], Seongjoon Park[1], Yuka Okada[9], Rina Takahashi[10], Yayoi Kawano[10], Takehisa Hanawa[10], Keisuke Iwasaki[11], Tadashige Nozaki[12], Hidetaka Torigoe[13], Kazuya Ikematsu[4], Yutaka Suzuki[6], Katsumi Tanaka[2], Paul Martin[7] [ID], Isao Shimokawa[1] & Ryoichi Mori[1,*] [ID]

## Abstract

Argonaute 2 bound mature microRNA (Ago2-miRNA) complexes are key regulators of the wound inflammatory response and function in the translational processing of target mRNAs. In this study, we identified four wound inflammation-related Ago2-miRNAs (*miR-139-5p*, *miR-142-3p*, *miR-142-5p*, and *miR-223*) and show that *miR-223* is critical for infection control. *miR-223*$^{Y/-}$ mice exhibited delayed sterile healing with prolonged neutrophil activation and interleukin-6 expression, and markedly improved repair of *Staphylococcus aureus*-infected wounds. We also showed that the expression of *miR-223* was regulated by CCAAT/enhancer binding protein alpha in human neutrophils after exposure to *S. aureus* peptides. Treatment with *miR-223*$^{Y/-}$-derived neutrophils, or *miR-223* antisense oligodeoxynucleotides in *S. aureus*-infected wild-type wounds markedly improved the healing of these otherwise chronic, slow healing wounds. This study reveals how *miR-223* regulates the bactericidal capacity of neutrophils at wound sites and indicates that targeting *miR-223* might be of therapeutic benefit for infected wounds in the clinic.

**Keywords** inflammation; *miR-223*; neutrophil; skin wound healing; *Staphylococcus aureus*

**Subject Categories** Immunology; Skin

See also: **P Hiebert & S Werner** (September 2018)

## Introduction

After tissue damage to the adult skin, there always follows a robust recruitment of inflammatory cells, including innate immune cells, neutrophils, and macrophages, into the wounded tissues to kill and phagocytose invading microbes as well as secreting bioactive substances that aid tissue repair (Eming *et al*, 2017). Healing of the damaged tissue is accomplished by the concerted actions of re-epithelialization and wound angiogenesis and by migration and subsequent contraction of fibroblasts that lay down granulation tissue by deposition of collagen and other matrix components (Eming *et al*, 2014). These contributing cell behaviors are thought to be partly governed by signals from the inflammatory cell influx (Eming *et al*, 2007). Following collagen deposition, a scar develops at the healed wound site (Eming *et al*, 2014). Interestingly, skin

---

1   Department of Pathology, Nagasaki University School of Medicine and Graduate School of Biomedical Sciences, Nagasaki, Japan
2   Department of Plastic and Reconstructive Surgery, Nagasaki University School of Medicine and Graduate School of Biomedical Sciences, Nagasaki, Japan
3   Department of Plastic and Reconstructive Surgery, Ehime Prefectural Center Hospital, Ehime, Japan
4   Department of Forensic Pathology and Science, Nagasaki University School of Medicine and Graduate School of Biomedical Sciences, Nagasaki, Japan
5   Department of Immunology and Rheumatology, Nagasaki University School of Medicine and Graduate School of Biomedical Sciences, Nagasaki, Japan
6   Laboratory of Functional Genomics, Department of Medical Genome Science, Graduate of Frontier Science, The University of Tokyo, Tokyo, Japan
7   Schools of Biochemistry and Physiology, Pharmacology & Neuroscience, Faculty of Biomedical Sciences, University of Bristol, Bristol, UK
8   Randall Division of Cell and Molecular Biophysics, King's College London, London, UK
9   Department of Ophthalmology, Wakayama Medical University, Wakayama, Japan
10  Faculty of Pharmaceutical Sciences, Tokyo University of Science, Chiba, Japan
11  Department of Pathology, Sasebo City General Hospital, Sasebo, Nagasaki, Japan
12  Department of Pharmacology, Faculty of Dentistry, Osaka Dental University, Hirakata, Osaka, Japan
13  Department of Applied Chemistry, Faculty of Science, Tokyo University of Science, Tokyo, Japan
    *Corresponding author. Tel: +81 95 819 7051; Fax: +81 95 819 7052; E-mail: ryoichi@nagasaki-u.ac.jp

---

wounds of embryos and late gestation fetuses (before embryonic day 15 in mice and end of second trimester in humans) can result in almost perfect repair without scarring and these early wounds are associated with a markedly reduced wound inflammatory response (Hopkinson-Woolley *et al*, 1994), suggesting that inflammation might cause skin fibrosis (Martin & Leibovich, 2005). Indeed, neonatal PU.1-deficient ($PU.1^{-/-}$) mice, which possess no neutrophils, macrophages, mast cells, or T cells, exhibit rapid repair and scarless healing in contrast to wild-type (WT) siblings (Martin *et al*, 2003; Cooper *et al*, 2005). We previously showed that knockdown of osteopontin, a wound upregulated inflammation-dependent gene, by antisense oligodeoxynucleotides (AS ODNs) at wound sites reduced scarring and improved healing *in vivo* (Mori *et al*, 2008). A similar effect was observed for wounds by knockdown of the transcription factor *Foxo1* (Mori *et al*, 2014). In general, these findings suggest that gene therapies based on dampening wound inflammatory responses (reducing inflammatory cell influx, blocking profibrotic signals from inflammatory cells once they arrive at the wound site, or enhancing the resolution of inflammation; Cash *et al*, 2014) might provide novel molecular therapeutic targets.

MicroRNAs (miRNAs) of the small, non-coding RNA family are approximately 20–25 nucleotides of single-stranded RNA that regulate translational processing of target mRNAs (Winter *et al*, 2009). miRNAs are degraded by selective loading into an RNA-induced silencing complex (RISC) with Argonaute (Ago) as its core. Ago family members (Ago1-4) are ubiquitously expressed, but Ago2 is the most highly expressed and exhibits miRNA silencer activity in mice (Liu *et al*, 2004). Many studies have now indicated that a number of diverse miRNAs are involved with, and regulate, inflammatory responses in humans and mice (O'Connell *et al*, 2012). Furthermore, the specific miRNAs *miR-21* (Wang *et al*, 2012), *miR-130a* (Pastar *et al*, 2012), and *miR-132* (Li *et al*, 2015) contribute to the healing of skin wounds. miRNA profiling using next-generation sequencing (NGS) has already proved successful in identifying novel miRNAs from several tissues and organisms (Tam *et al*, 2014; Ma *et al*, 2015). However, methods to purify Ago2-miRNA complexes from wound sites and miRNA library construction to perform NGS from these tissues have not previously been reported.

*miR-223*$^{Y/-}$ mice have an expanded granulocytic compartment resulting from a cell-autonomous increase in the number of granulocyte progenitors (Johnnidis *et al*, 2008). Moreover, *miR-223* is overexpressed in rheumatoid arthritis patients (Fulci *et al*, 2010). Together, these studies suggest that inflammation-related miRNAs, particularly *miR-223*, might be key regulators of the inflammatory response and/or its subsequent resolution during skin tissue repair; however, the pathways involved have not been comprehensively characterized.

In this study, we developed a unique purification system to isolate functional miRNA-Ago2 complexes from wounded skin tissues by immunoprecipitation (IP) using an anti-Ago2 antibody (Ab), followed by the construction of libraries to perform NGS and identify candidate wound inflammation-related miRNAs. Using this approach, we identified several inflammation-dependent miRNAs including *miR-139-5p*, *miR-142-3p*, *miR-142-5p*, and *miR-223*. Of these, *miR-223* was the most highly expressed in wound sites during the inflammatory phase and so the present study focused on the molecular mechanisms of *miR-223* in skin wound healing. Our wound healing studies showed the delayed repair of aseptic wounds

in *miR-223*$^{Y/-}$ mice was associated with enhanced neutrophil activation and interleukin-6 (*Il6*) expression. However, if wounds were infected with *Staphylococcus aureus*, then *miR-223*$^{Y/-}$ mice showed considerably enhanced repair compared with WT infected wounds, and either transplanting *miR-223*$^{Y/-}$ neutrophils or the delivery of *miR-223* AS ODNs to infected WT wounds rescued the impaired wound healing phenotype. The expression level of *miR-223* in human neutrophils was regulated by CCAAT/enhancer binding protein alpha (C/EBPα) after exposure to *S. aureus* peptides. Thus, *miR-223* is a potential therapeutic target for the treatment of skin wounds infected with *S. aureus*.

# Results

## Over 300 Ago2-miRNAs are expressed during skin wound healing

To comprehensively identify wound-induced mature miRNAs involved in skin wound healing, a 4-mm-diameter wound was made in WT mouse dorsal skin, and 6-mm-diameter unwounded and wound sites were harvested on day 1, 3, 7, 10, and 14 after injury as described previously (Mori *et al*, 2008, 2014). Because mature Ago2-miRNA complexes bind to target mRNAs and inhibit their translation, we isolated Ago2-miRNA complexes rather than total miRNAs. We purified Ago2-miRNA complexes from skin wound tissues by IP with Ago2 Ab, followed by library generation and NGS with the Illumina platform (Appendix Fig S1). Known murine miRNA sequencing reads accounted for 76.6–91.0% of all sequence reads in our small RNA libraries that were highly enriched for mature miRNA sequences (Appendix Table S1). We identified over 300 known murine miRNA categories expressed during skin wound healing (Dataset EV1), demonstrating the high efficiency of our NGS procedure for identifying functional miRNAs from skin wound tissues.

## *miR-139-5p*, *miR-142* family members, and *miR-223* are wound inflammation-related miRNAs

As a partial filter to identify inflammation-related miRNAs in skin wound healing, miRNAs in the inflammatory phase (day 1 after injury) were arranged by rank of upregulation and compared with intact skin (Appendix Table S2). miRNA expression levels were confirmed by qPCR (Fig 1). *miR-147*, *miR-223* (*miR-223-3p*), *miR-129-3p*, *miR-129-5p*, *miR-139-5p*, *miR-21\** (*miR-21-3p*), *miR-340-5p*, *miR-142-3p*, and *miR-142-5p* expressions at wound sites on day 1 after injury were significantly increased compared with intact skin.

In WT neonatal mice, *miR-223* and *miR-142-3p* expression levels were markedly increased, peaking at 12 h after injury compared with intact skin, and were subsequently significantly decreased by 24 h (Fig 2A) indicating a temporal association with the inflammatory phase of healing. To confirm that these were inflammation-related miRNAs expressed during skin wound healing, we made a 1-cm incisional wound in the dorsal skin of WT neonatal mice and compared miR expression with heterozygous PU.1-deficient ($PU.1^{+/-}$) and $PU.1^{-/-}$ mice. qPCR analysis showed *miR-223*, *miR-142-3p*, *miR-142-5p*, and *miR-139-5p* expression levels at wound sites in $PU.1^{-/-}$ mice were significantly decreased compared with WT mice (Fig 2B), indicating that these four miRs are inflammation-related miRNAs expressed during skin wound healing.

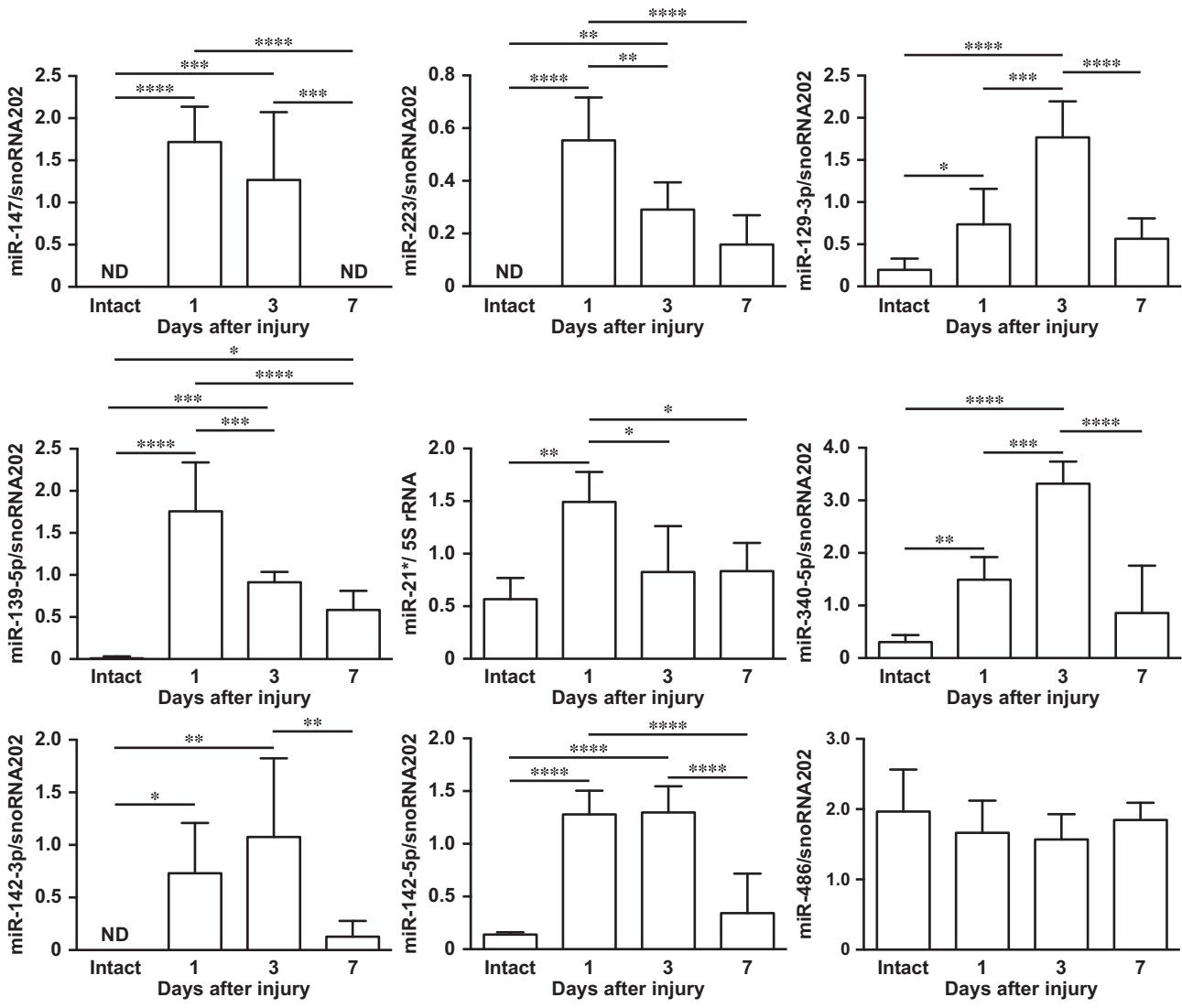

**Figure 1.   Identification and expression of nine candidate inflammation-related miRNAs in skin wound healing.**

Temporal expression of murine *miR-147*, *miR-223*, *miR-129-3p*, *miR-139-5p*, *miR-21\**, *miR-340-5p*, *miR-142-3p*, *miR-142-5p*, and *miR-486* in skin wound healing measured by qPCR relative to snoRNA202 or 5S rRNA ($n = 4$–6). All values represent the mean $\pm$ SD. Tukey's multiple comparison tests were used to generate the *P*-values indicated in the Figure. \**P* < 0.05, \*\**P* < 0.01, \*\*\**P* < 0.001, \*\*\*\**P* < 0.0001.

## Skin wound healing is delayed in *miR-223*$^{Y/-}$ mice

Of all inflammation-related miRNAs, *miR-223* was the most highly expressed during the inflammatory phase at wound sites and so the present study focused on the molecular mechanisms of *miR-223* in skin wound healing (Appendix Fig S2). To determine which cells expressed *miR-223* during skin wound healing, we performed immunohistochemistry (IHC) and *in situ* hybridization (ISH). Wound-infiltrated neutrophils at day 1 in the wound site of WT mice predominantly expressed *miR-223* (Fig 3A and B), and this was similar in human skin-inflamed sites (Fig 3C). We used qPCR to confirm that murine wound-infiltrating neutrophils (1 day after wounding) and macrophages (3 days after wounding) isolated by immunoaffinity selection with anti-Ly-6G (neutrophil marker) and anti-CD11b

(macrophage marker) Abs (Tanaka *et al*, 2017) expressed *miR-223* at the wound sites (Fig 3D). Thus, *miR-223* is predominantly expressed in neutrophils and macrophages at skin wound sites.

To clarify the pathophysiological role of *miR-223* in skin wound healing, we made dorsal skin wounds in *miR-223*$^{Y/-}$ mice. Gross examination showed that wound closure was significantly delayed in *miR-223*$^{Y/-}$ mice compared with WT mice (Fig 3E and F). We investigated re-epithelialization using histological analysis and observed that epithelial wound tongues in *miR-223*$^{Y/-}$ mice at day 3 after injury were significantly shorter (188.3 $\pm$ 24.11 μm, $P = 0.0115$) than in WT mice (380.1 $\pm$ 47.08 μm; Fig 3G and H) confirming that *miR-223*$^{Y/-}$ mice showed delayed wound re-epithelialization.

We investigated the area of granulation tissue at day 7 and day 14 at wound sites in *miR-223*$^{Y/-}$ mice and found them to be

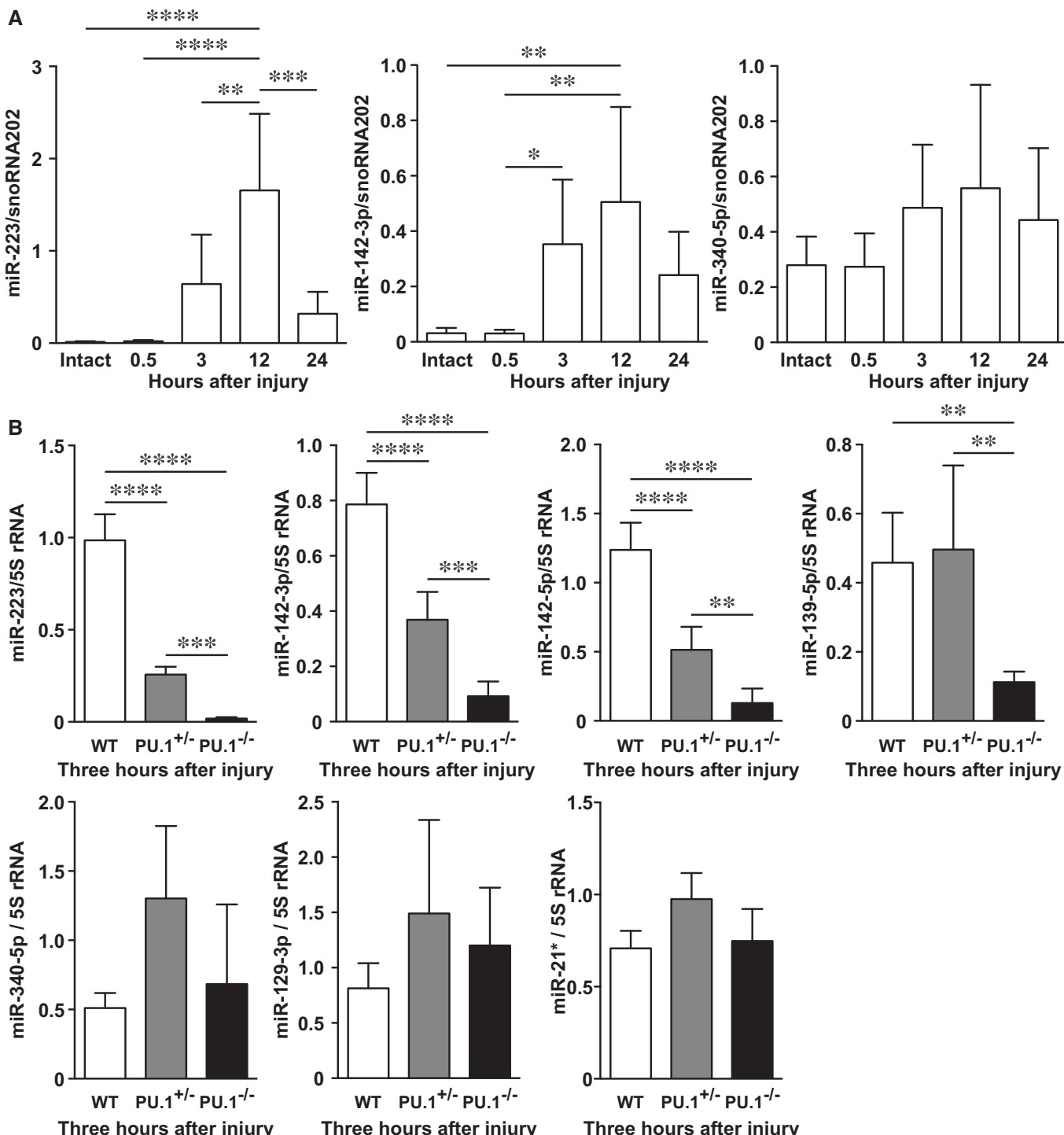

**Figure 2. *miR-139-5p*, *miR-142-3p*, *miR-142-5p*, and *miR-223* are independently expressed by wound inflammation.**

A   Temporal expression of *miR-223*, *miR-142-3p*, and *miR-340-5p* in skin wound healing of neonate mice using qPCR ($n = 6$).

B   qPCR analysis indicates the expression levels of inflammation-related miRNAs at 3 h in wound sites of WT ($n = 5$), *PU.1⁺/⁻* ($n = 6$), and *PU.1⁻/⁻* ($n = 6$) neonatal mice.

Data information: Results shown are the mean ± SD. Tukey's multiple comparison tests were used to generate the *P*-values indicated in the Figure. *$P < 0.05$, **$P < 0.01$, ***$P < 0.001$, ****$P < 0.0001$.

significantly increased (day 7; 0.60 ± 0.16 mm², day 14; 0.18 ± 0.064 mm²) compared with WT mice (day 7; 0.36 ± 0.16 mm², day 14; 0.11 ± 0.036 mm²; Fig EV1A–C). We

examined the localization and expression level of α-smooth muscle actin (αSMA), a marker of contractile myofibroblasts (Gabbiani *et al*, 1971), using IHC. The expression of αSMA at day 7 in

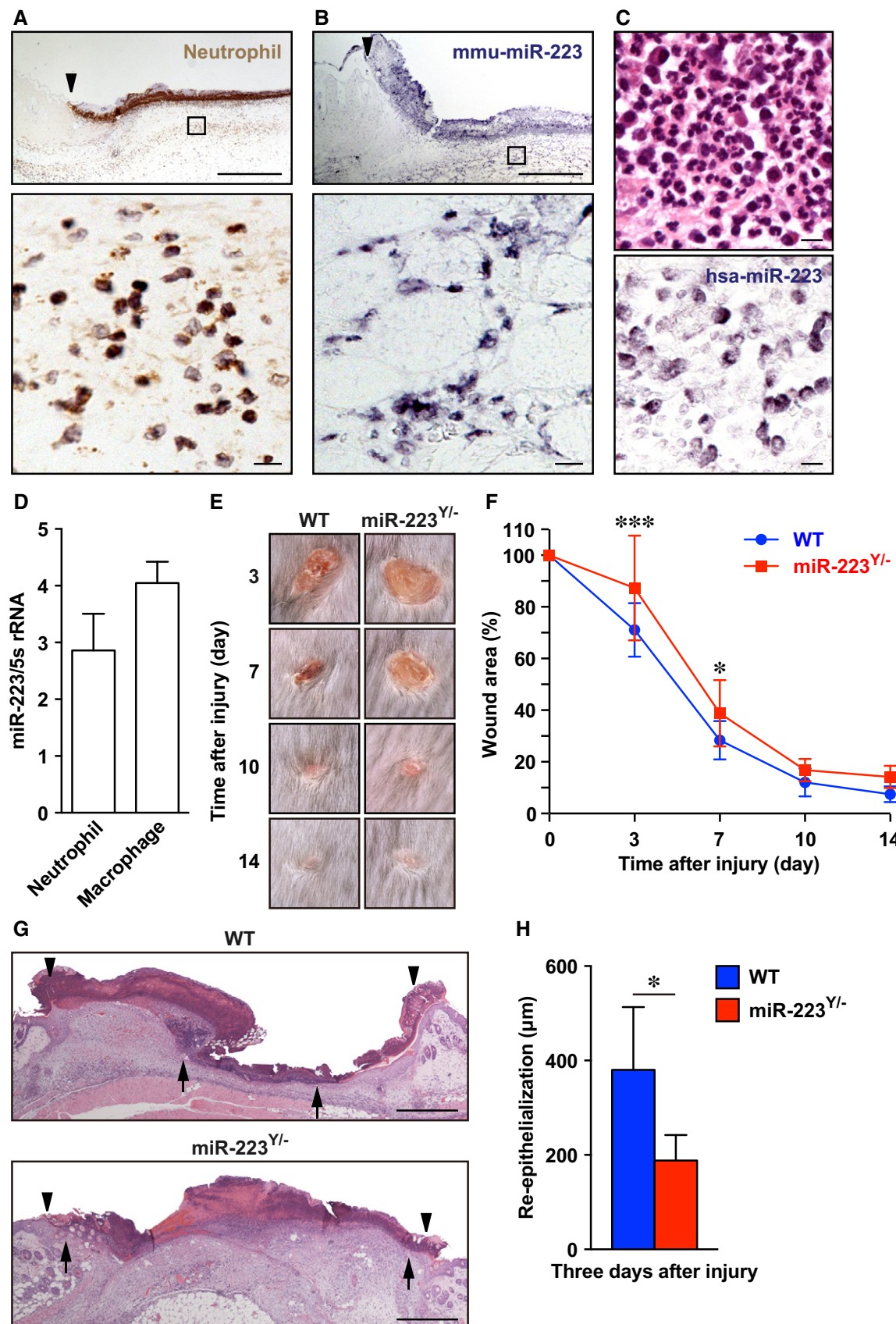

**Figure 3.**

**Figure 3. *miR-223*^Y/− mice show delayed aseptic skin wound healing.**

A   IHC of neutrophils (Ly-6G and Ly-6C) at day 1 wound sites. Arrowhead indicates the wound margin. Area indicated with a rectangle is shown at a higher magnification below. Scale bars: 50 μm (bottom) and 500 μm (top).

B   ISH of *Mus musculus* (mmu) *miR-223* showing that wound-infiltrated neutrophils were predominantly present in the wound sites of adult WT mice at day 1 after injury. Arrowhead indicates the wound margin. Area indicated with a rectangle is shown at a higher magnification below. Scale bars: 50 μm (bottom) and 500 μm (top).

C   Neutrophils express *Homo sapiens* (hsa) *miR-223* in human skin-inflamed sites. Representative H&E staining (top) and ISH of *miR-223* (bottom) showing that *miR-223*-expressing neutrophils were predominantly present in patient skin-inflamed sites (*n* = 15, Appendix Table S3). Scale bars: 50 μm.

D   qPCR analysis shows *miR-223* was expressed by wound-infiltrating neutrophils (day 1) and macrophages (day 3) (*n* = 3).

E   Representative photo images of the gross appearance of excisional wounds in WT (left) and *miR-223*^Y/− (right) mice.

F   Proportion of the wound area remaining open relative to initial wound area at each time point in WT (*n* = 14) and *miR-223*^Y/− (*n* = 13) mice.

G   H&E staining of re-epithelialization at day 3 after injury (wound margin (arrowheads) and the leading edge of epithelia (arrows)). Scale bar: 500 μm.

H   Measurement of epithelial tongue at day 3 after injury in WT (*n* = 8) and *miR-223*^Y/− (*n* = 5) mice.

Data information: All values represent the mean ± SD. Two-way ANOVA followed by Sidak multiple comparisons test (F) and unpaired *t*-test (H) was used to generate *P*-values indicated in the Figure. **P* < 0.05, ****P* < 0.001.

granulation tissues of *miR-223*^Y/− mice was markedly decreased (58%) compared with WT mice, even though the localization of αSMA-expressing cells at day 7 in granulation tissues of *miR-223*^Y/− mice was not altered when compared with WT mice (Fig EV1D and E). Collectively, *miR-223*^Y/− mice showed delayed skin wound healing and an increased scar area.

### *MiR-223* regulates neutrophils in the acute inflammatory responses at wound sites

Because we found *miR-223* was expressed in neutrophils at wound sites and *miR-223*^Y/− mice exhibited significantly delayed skin repair, we investigated whether *miR-223* regulates neutrophil functions in acute inflammatory responses, in ways which might impact on the recruitment of macrophages leading to a further chronic inflammatory response. To investigate how *miR-223* might regulate the duration of the inflammatory response during skin wound healing *in vivo*, we examined neutrophil migration into wound sites during repair using lysozyme M (*lys*)-enhanced green fluorescent protein (*lys*-EGFP) mice, in which most myelomonocytic cells, especially mature neutrophil granulocytes, specifically express EGFP (Faust *et al*, 2000). *In vivo* live imaging analysis using these mice allowed semi-quantifiable evaluation of the spatiotemporal recruitment of neutrophils to skin wound sites, because EGFP fluorescence intensity is directly correlated with the number of neutrophils at wound sites (Kim *et al*, 2008; Tanaka *et al*, 2017). At 3 h, neutrophils began to accumulate at the margins of wounds in WT:*lys*-EGFP

mice (by measuring EGFP fluorescence intensity) with the intensity peaking at day 3 after injury (Fig 4A and B). In contrast, no migration or accumulation of neutrophils was observed in *miR-223*^Y/−:*lys*-EGFP mice at 3 h after injury; however, after 12 h the number of infiltrated neutrophils was greater than that of the WT neutrophil influx. At day 3, the number of neutrophils measured by the intensity of EGFP in *miR-223*^Y/−:*lys*-EGFP mice was significantly increased compared with WT:*lys*-EGFP mice.

We next examined the function of neutrophils *in vivo* as an index of myeloperoxidase (MPO) activation, a marker for neutrophil activation at wound sites (Mori *et al*, 2014). *In vivo* live imaging analysis using luminescence was used to monitor dynamic changes in neutrophil activation at skin wound sites. We found that MPO activation was markedly increased at wound sites of *miR-223*^Y/− mice compared with WT mice at day 1 and day 3 after injury, which correlated with the amount of MPO at skin wound sites (Fig 4C–E).

2-[6-(4′-amino)phenoxy-3H-xanthen-3-on-9-yl]benzoic acid (APF) is a fluorescent probe that selectively binds to reactive oxygen species (ROS) such as hydroxyl radicals, peroxynitrite, and hypochlorite produced by MPO in neutrophils (Setsukinai *et al*, 2003). To dissect the time-dependent activation of neutrophils, we performed live cell imaging analysis using confocal microscopy. We observed that ROS production in peripheral blood neutrophils (PBNs) derived from *miR-223*^Y/− mice was markedly delayed after incubation with phorbol myristate acetate (PMA) for 30 min (Fig 4F and G; Movies EV1 and EV2). Interestingly, *miR-223*^Y/− PBNs subsequently exhibited marked hyperactivation at 60 min compared with

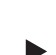

**Figure 4. *miR-223* regulates neutrophil activation at wound sites.**

A   Representative *in vivo* fluorescent images of EGFP-expressing neutrophils at skin wound sites. Active fluorescent imaging on color scale overlaid on a gray scale image of wounds sites.

B   Recruitment of neutrophils at skin wound sites measured by *in vivo* fluorescence. All values indicate the mean radiant efficiency (photon/sec/cm$^2$/sr)/(uW/cm$^2$) (*n* = 3).

C   Representative *in vivo* MPO activity bioluminescence imaging on color scale overlaid on a gray scale image of wound sites.

D   Measurements of MPO activity *in vivo* at wound sites of WT and *miR-223*^Y/− mice. All values are the mean surface radians (photon/sec/cm$^2$/sr) (day 1; *n* = 9, day 3; *n* = 12).

E   MPO concentrations measured by ELISA revealed MPO levels of *miR-223*^Y/− mice were significantly increased compared with WT mice (intact skin; *n* = 4, day 1; *n* = 5 (*miR-223*^Y/−) or *n* = 8 (WT), day 3; *n* = 6 (*miR-223*^Y/−) or *n* = 5 (WT)).

F   Representative live *in vitro* fluorescence imaging of ROS production in WT and *miR-223*^Y/−-derived neutrophils (Movies EV1 and EV2). Scale bar: 10 μm.

G   Temporal ROS production in neutrophils from WT (32 neutrophils from 3 mice, blue) and *miR-223*^Y/− (36 neutrophils from 3 mice, red) mice were measured at 60 min after PMA stimulus.

Data information: Values represent the mean ± SD (B, D, and E), or ± SEM (G). Two-way ANOVA followed by Sidak multiple test (B), unpaired *t*-test (B, D, and E), and Mann–Whitney *U*-test (G) were used to generate *P*-values indicated in the Figure. **P* < 0.05.

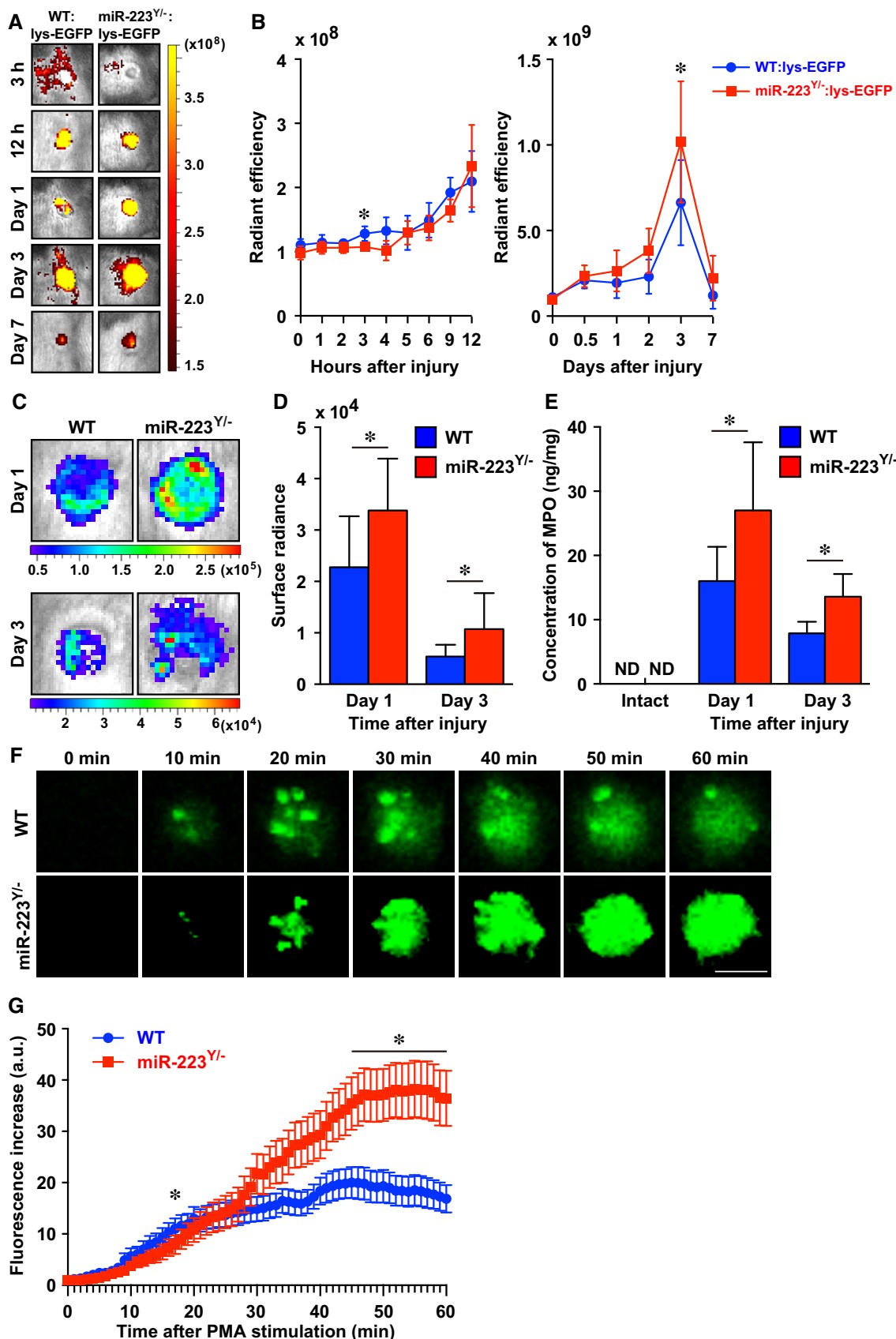

**Figure 4.**

WT PBNs (14.6 ± 2.4 in WT PBNs versus 26.1 ± 4.3 in *miR-223*[Y/−] PBNs, *P* = 0.0113), suggesting *miR-223* might be involved in neutrophil activation.

Macrophages infiltrate wound sites in the later inflammatory phase after neutrophil migration (Eming *et al*, 2014). IHC for F4/80, a macrophage marker (Austyn & Gordon, 1981), revealed that macrophage numbers were significantly increased in the wound sites of *miR-223*[Y/−] mice at day 3 and 7 after injury (Fig EV2A and B). In agreement with the altered macrophage infiltration time course, macrophage inflammatory protein (MIP)-1α was significantly increased at day 3 after injury in the wound sites of *miR-223*[Y/−] mice (Fig EV2C). Collectively, our *in vivo* and *in vitro* analyses indicate that *miR-223* might regulate the acute inflammatory response at wound sites and subsequently affect macrophage infiltration into wound sites.

### *miR-223* regulates *Il6* expression at wound sites

We investigated the molecular mechanisms of transcriptome regulation by *miR-223* at wound sites using NGS mRNA-Sequencing (mRNA-Seq) of day 1 skin wound samples from *miR-223*[Y/−] and WT mice (Dataset EV2). We screened potential *miR-223* target mRNAs using bioinformatics and our NGS data using a fold-change cutoff of 1.5 (Appendix Fig S3). We identified 2 *miR-223* target mRNAs, *Il6* (WT: 13.3, *miR-223*[Y/−]: 31.3; 2.36-fold increase in expression level) and Piccolo (*Pclo*), a presynaptic cytomatrix protein (WT: 0.025, *miR-223*[Y/−]: 0.043, 1.71-fold increase). *Il6* is a classic proinflammatory cytokine gene, and IL-6 protein is produced locally at human skin wound sites 30 min after injury (Grellner *et al*, 2000). IHC revealed that wound-infiltrating neutrophils predominantly expressed and secreted IL-6 in WT and *miR-223*[Y/−] wound sites at day 1 (Fig 5A) and was significantly overexpressed in *miR-223*[Y/−] wound sites at day 1 compared with WT mice (Fig 5B). However, expression levels of IL-6 receptor-α (*Il6ra*), an IL-6 signal transducer (*Il6st*) that is also a candidate *miR-223* target mRNA, and the proinflammatory cytokine gene *Il1b* were not altered (Fig 5C–E). Interestingly, non-stimulated bone marrow-derived *miR-223*[Y/−] neutrophils, but not WT neutrophils, constitutively secreted IL-6 *in vitro* (Fig 5F). To determine whether *miR-223* seed sequences dependently bind to *Il6* target mRNAs in cells, we designed three *miR-223* point mutation mimics (Fig 5G). The *miR-223* mimic bound to the *Il6* 3′-untranslated regions (UTRs) and was seed sequence dependent compared with the control and two *miR-223* point mutation mimics with 2 and 4 point mutations in the seed sequence, respectively (Fig 5H). These data suggest that *miR-223*

might directly regulate IL-6 translation and secretion from neutrophils at wound sites.

### *miR-223*[Y/−] neutrophils contribute to improved healing of *Staphylococcus aureus*-infected wounds

*Staphylococcus aureus* is the most common cause of skin infection in humans and is frequently detected in severe chronic skin wound sites in the clinic (Salgado-Pabon & Schlievert, 2014). Long-term *S. aureus* infection can lead to methicillin-resistant *S. aureus*. Our studies showed that *miR-223*[Y/−] neutrophils were hyperactivated and because they expressed IL-6, which was important for *S. aureus* clearance in a subcutaneous infection model using *Il6*[−/−] mice (Hruz *et al*, 2009), we investigated whether *miR-223*[Y/−] neutrophils preferentially kill bacteria at wound sites. Wounds in WT and *miR-223*[Y/−] mice were inoculated with *S. aureus* ($1 \times 10^8$ colony-forming units [CFU]/10 μl), which led to impaired wound healing in WT mice; these wounds were still not healed by day 7 after injury, with overt healing only initiated at day 10 to day 14. By contrast, *miR-223*[Y/−] mice showed significantly enhanced healing of *S. aureus*-infected wounds at day 3 and 7 compared with controls (Fig 6A and B), accompanied by enhanced re-epithelialization, reduced total wound area and pathological postinfectious necrotic lesions without significant alteration in area of granulation tissue, and the expression of αSMA in granulation tissues at day 7, but not at day 14 (Fig EV3). However, the percentage of granulation tissue area within the total wound area of *miR-223*[Y/−] mice at day 7 was significantly increased (72.2% ± 23.1%, *P* = 0.0494) compared with WT mice (35.4% ± 33.1). The colonization of *S. aureus* at day 3 in the wound sites of *miR-223*[Y/−] mice was significantly reduced ($3.67 \times 10^4$ ± 7149 CFU/wound) compared with WT mice ($1.16 \times 10^5$ ± 32,849 CFU/wound; Fig 6C).

Abscess formation is a defensive reaction of tissues to prevent the spread of infecting bacteria to other sites and previous studies showed it was indispensable for promoting healing in *S. aureus*-infected wounds (Kobayashi *et al*, 2015). Our histological observations demonstrated a significantly increased abscess size in day 1 wounds in *miR-223*[Y/−] mice (0.25 ± 0.04 mm$^2$) compared with WT mice (0.13 ± 0.04 mm$^2$; Fig 6D and E).

We also investigated the expression levels of neutrophil-derived *Il6* and *Il1b* that are positively linked to *S. aureus* clearance (Hruz *et al*, 2009; Cho *et al*, 2012) by qPCR, and found that they were both markedly increased in *S. aureus*-infected wound sites of *miR-223*[Y/−] mice at day 3 compared with equivalent wound sites of WT mice (Fig 6F).

---

**Figure 5. Neutrophil IL-6 secretion is regulated by *miR-223* at wound sites.**

A   IHC for IL-6 (red) and neutrophils (green) shows neutrophils express IL-6 at day 1 after injury in WT and *miR-223*[Y/−] mice. Nuclei were counterstained with DAPI. Secreted IL-6 was deposited around neutrophils and in wound sites. Scale bar: 10 μm.

B–E   qPCR analysis indicates the expression levels of *Il6* (B), *Il6ra* (C), *Il6st* (D), and *Il1b* (E) at day 1 in wound sites of WT and *miR-223*[Y/−] mice (*n* = 6).

F   ELISA reveals non-stimulated bone marrow-derived *miR-223*[Y/−] neutrophils constitutively secreted IL-6 (*n* = 3). Non-stimulated bone marrow-derived WT neutrophils were not detected.

G   Alignment of seven *miR-223* seed sequences and the corresponding seed sequences in *Il6* mRNAs (red color and capital letters, respectively). Point mutations in each mutation mimic are shown as green, brown, and blue, respectively. The digit indicates the 3′-UTR position.

H   *miR-223* binds to *Il6* 3′-UTRs with high affinity. A luciferase reporter vector encoding 3′-UTRs was co-transfected with *miR-223* mimics into 3T3 cells. A decrease in luciferase activity indicates binding of the *miR-223* mimic to the 3′-UTR of the target sequence (*n* = 6–8).

Data information: All values represent the mean ± SD. Unpaired Student's *t*-test (B) and ordinary one-way ANOVA followed by Dunnett's multiple comparisons test (*miR-223* mimic vs each sample) (H) were used to generate *P*-values indicated in the Figure. **P* < 0.05, ***P* < 0.01.

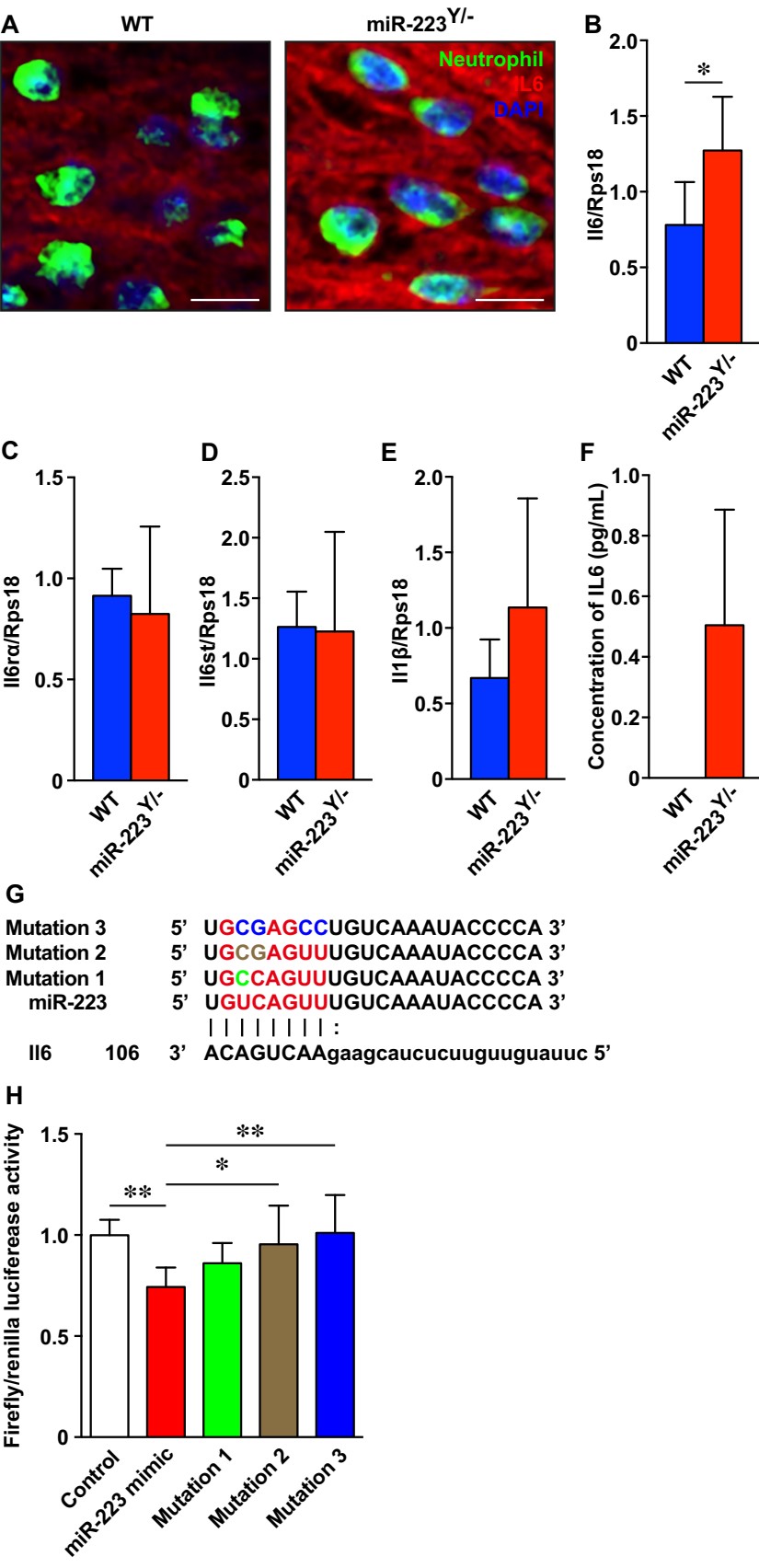

**Figure 5.**

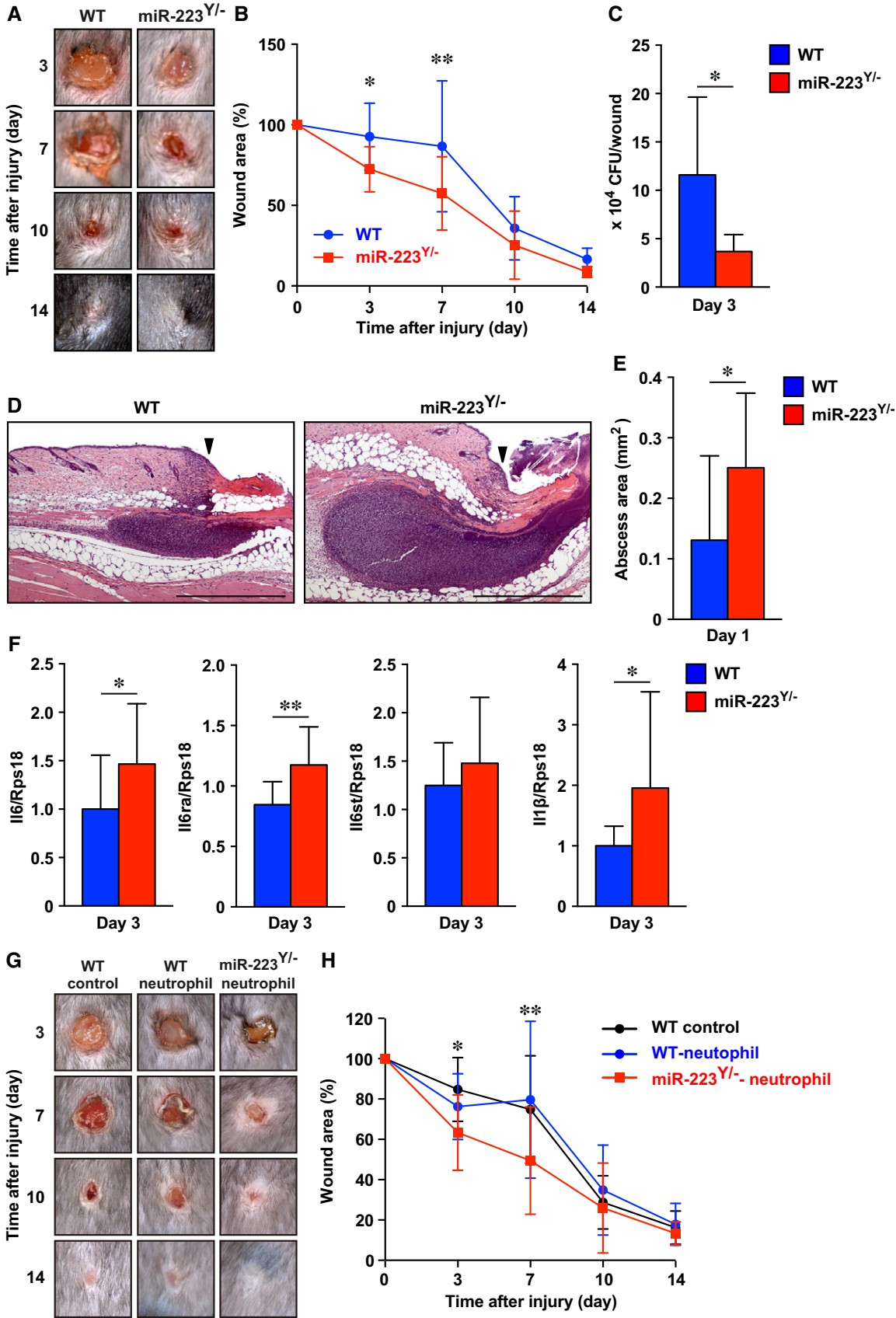

Figure 6.

◀

**Figure 6.  miR-223^{Y/−} neutrophils improve *Staphylococcus aureus*-infected skin wound healing.**

A   Representative photo images of the gross appearance of *S. aureus*-infected wounds in WT (left) and *miR-223*^{Y/−} (right) mice.
B   Proportion of the wound area remaining open relative to initial wound area at each time point in *S. aureus*-infected WT (*n* = 11, blue) and *miR-223*^{Y/−} (*n* = 13, red) mice.
C   Amount of *S. aureus* in wound sites of *miR-223*^{Y/−} at day 3 were significantly decreased (*n* = 6).
D   Representative images of H&E staining of abscess formation at day 1 wound sites. Arrowheads indicate wound margin. Scale bars: 500 μm.
E   Measurement of abscess area of 1 day infected wounds (WT; *n* = 11, *miR-223*^{Y/−}; *n* = 12).
F   qPCR analysis showed *Il6*, *Il6ra*, and *Il1b* expressions in *S. aureus*-infected wound sites of *miR-223*^{Y/−} mice were significantly increased (WT; *n* = 16, *miR-223*^{Y/−}; *n* = 15).
G   Representative photo images of the gross appearance of non-transplant WT control (left), WT neutrophil-transplanted (middle), and *miR-223*^{Y/−} neutrophil-transplanted (right) wounds.
H   Proportion of the wound area remaining open relative to initial wound area at each time point. Note that wound closure of *miR-223*^{Y/−} neutrophil-transplanted sites was significantly faster than non-transplant and WT neutrophil-transplanted wounds (WT control *n* = 11, WT neutrophil *n* = 12, *miR-223*^{Y/−} neutrophil *n* = 13).

Data information: All values represent the mean ± SD. Unpaired *t*-tests (E and F (*Il6* and *Il6ra*)) followed by Welch's test (C and F (*Il1b*)) and two-way ANOVA followed by Sidak multiple test (B) and Tukey's multiple test (H) were used to generate the *P*-values indicated in the Figure. \**P* < 0.05, \*\**P* < 0.01.

Our studies described above suggest the potential beneficial effects of cell transplantation therapy using *miR-223*^{Y/−} neutrophils engrafted in *S. aureus*-infected wound sites. To test the feasibility of such a therapy, we applied bone marrow-derived neutrophils (1 × 10^6 neutrophils in 50 μl of saline/wound, number of neutrophils was 1:100) to *S. aureus*-infected WT skin wound sites. We confirmed that transplanted WT neutrophils remained in *S. aureus*-infected skin wound sites up to 3 days after transplantation (Appendix Fig S4A). *miR-223*^{Y/−} neutrophil-treated skin wound sites showed significantly improved healing accompanied by a reduced total wound area and increased the expression of αSMA in granulation tissues at day 7 and a reduced area of granulation tissue at 7 and 14 days when compared with non-treated or WT neutrophil-treated *S. aureus*-infected wound sites (Fig 6G and H; Appendix Fig S4B–E). This indicated that *miR-223*^{Y/−} neutrophils might have therapeutic benefit for *S. aureus*-infected skin wound healing.

**Acute knockdown of *miR-223* using AS ODNs at *Staphylococcus aureus*-infected wounds sites improves healing**

Our data provide experimental evidence that blocking *miR-223* activity may improve *S. aureus*-infected skin wound healing. Because molecular targeting by the delivery of AS ODN to wounds knocked down specific mRNAs at skin wound sites (Mori *et al*, 2006, 2008, 2014), we wondered whether a similar approach might

knockdown *miR-223* expression. To test this hypothesis and investigate whether the acute knockdown of *miR-223* activity might be a feasible therapeutic strategy for improving the healing of infected wounds, we designed and optimized locked nucleic acid (LNA)-modified *miR-223*-specific AS ODN *in vitro* (Fig 7A and B).

We prepared a novel poloxamer P407/P188 binary thermosensitive hydrogel (PB gel) that is highly effective for ODN delivery to tissues. The characteristics of PB gel include its liquid form at room temperature but rapid solidification at body temperature, efficient release of 18-mer ODNs by 1 h (R. Takahashi *et al*, to be submitted elsewhere), and 30% increased efficiency compared with poloxamer Pluronic gel, which is currently a standard tool for ODN delivery to wound sites (Mori *et al*, 2006, 2008, 2014; Fig EV4A and B). We confirmed that the expression of *miR-223* in control ODN (with a sequence predicted to be non-binding to other mRNAs and miRNAs)-treated *S. aureus*-infected skin wound sites was not altered compared with control PB gel-treated *S. aureus*-infected skin wound sites (Fig 7C). We applied *miR-223*-specific AS ODN in PB gel versus control ODN in PB gel to *S. aureus*-infected skin wound sites and confirmed that the expression of *miR-223* in *miR-223* AS ODN-treated *S. aureus*-infected skin wound sites was significantly decreased (6 h, 0.022 ± 0.013; day 1, 1.16 ± 0.45) compared with control ODN-treated *S. aureus*-infected skin wound sites (6 h, 3.46 ± 1.10; day 1, 7.57 ± 1.12; Fig 7C and D). The expression of IL-6 at day 1 in *miR-223* AS ODN-treated *S. aureus*-infected skin

**Figure 7.  miR-223 AS ODN improves *Staphylococcus aureus*-infected skin wound healing.**

A   L, and ^ indicate LNA, and phosphorothioated ODN, respectively. Alignment of *miR-223* AS ODN sequences and the corresponding sequences in mature *miR-223*.
B   Confirmation of *miR-223* AS ODN binding activity *in vitro*. *miR-223* AS ODN binds to the mature *miR-223* region with a high affinity. A luciferase reporter vector encoding the mature *miR-223* was co-transfected with *miR-223* AS ODN into 3T3 cells. A decrease in luciferase activity indicates binding of the control ODN to the target sequence (*n* = 4).
C   Expression of *miR-223* in PB gel or control ODN-treated *S. aureus*-infected wound sites using qPCR (*n* = 4).
D   Temporary expression of *miR-223* in control ODN or *miR-223* AS ODN-treated *S. aureus*-infected wound sites using qPCR (*n* = 4).
E   Expression of *Il6* at day 1 in control ODN or *miR-223* AS ODN-treated *S. aureus*-infected wound sites using qPCR (*n* = 4).
F   Representative photo images of the gross appearances of control and *miR-223* AS ODN-treated wounds at various time points after wounding (*n* = 7).
G   The proportion of the wound area remaining open relative to the initial *S. aureus*-infected wound area at each time point after the injury in control ODN versus *miR-223* AS ODN-treated wounds (*n* = 7).
H   Measurement of total wound area, necrotic lesion area, and granulation tissue area at day 7 after injury in control ODN or *miR-223* AS ODN-treated *S. aureus*-infected wound sites (*n* = 6).
I   Quantification of the expression of αSMA at day 7 in control ODN (*n* = 5) or *miR-223* AS ODN-treated *S. aureus*-infected wound sites (*n* = 6).
J   Measurement of the area of granulation tissues at day 14 in control ODN or *miR-223* AS ODN-treated *S. aureus*-infected wound sites (*n* = 6)

Data information: All values represent the mean ± SD. Unpaired *t*-tests (B, D, E, I, and J) followed by Welch's correction (H) and two-way ANOVA followed by Sidak multiple test (G) were used to generate the *P*-values indicated in the Figure. \**P* < 0.05, \*\**P* < 0.01, \*\*\**P* < 0.001, \*\*\*\**P* < 0.0001.

▶

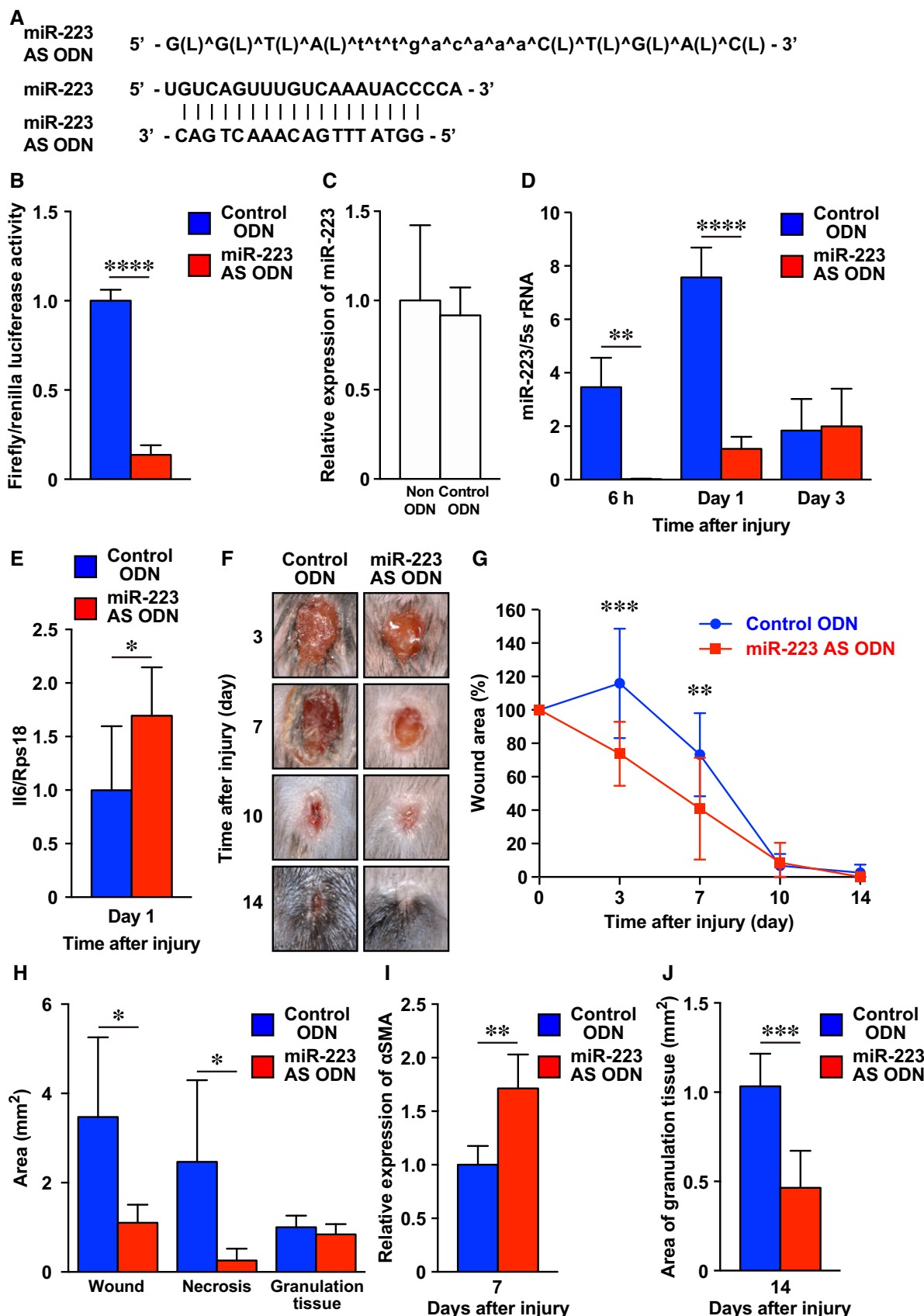

**Figure 7.**

wound sites was significantly increased compared with control ODN-treated *S. aureus*-infected skin wound sites (Fig 7E). Macroscopic analysis of wounds at day 3 and day 7 after injury indicated that wound closure in *miR-223* AS ODN-treated *S. aureus*-wound sites was markedly accelerated (day 3; $73 \pm 19\%$, day 7; $41 \pm 30\%$) compared with control ODN-treated *S. aureus*-infected wounds (day 3; $115 \pm 33\%$, day 7; $73 \pm 24\%$; Fig 7F and G). Histological analysis indicated that the total wound area and necrotic lesions in *miR-223* AS ODN-treated *S. aureus*-infected wound sites at day 7 were markedly decreased compared with controls and were accompanied by the enhanced expression of αSMA in granulation tissues (Fig 7H and I). The percentage of granulation tissue contribution to total wound area of *miR-223* AS ODN-treated *S. aureus*-infected wound sites was significantly increased ($80.9 \pm 17.7\%$, $P = 0.0017$) compared with control ODN-treated *S. aureus*-infected wound sites ($36.4\% \pm 18.6$). By 14 days post-wounding, the area of granulation tissue in *miR-223* AS ODN-treated *S. aureus*-infected wound sites was significantly reduced compared with controls (Fig 7J). Our data suggest that the acute downregulation of *miR-223* at wound sites using *miR-223* AS ODN accelerated skin healing in *S. aureus*-infected wounds as effectively as *miR-223*-deficient neutrophil transplantation, as described earlier.

### IL-6 secretion from *Staphylococcus aureus*-recognizing human neutrophils is regulated by C/EBPα-mediated *miR-223* expression

To clarify the molecular mechanisms of the *miR-223*/IL-6 secretion pathway in human neutrophils after infection with *S. aureus*, we established a *miR-223* knockdown human promyelocytic leukemia HL-60 cell line that could be differentiated to neutrophils upon treatment with DMSO using *miR-223* AS ODN. As expected, the expression of *miR-223* in DMSO-treated HL-60 cells increased as they progressed in their differentiation to mature neutrophils (Fig 8A) because *miR-223* contributed to neutrophil development (Johnnidis *et al*, 2008).

Next, we stimulated the differentiation HL-60 (dHL-60) cells with *S. aureus*-derived peptidoglycan (PGN), a ligand for Toll-like receptor (TLR)-2 (Takeuchi *et al*, 1999), and investigated the expression pattern of *miR-223* in neutrophils after the recognition of *S. aureus*. The expression of *miR-223* in dHL-60 cells stimulated with PGN at 6 h was significantly deceased (38%) compared with non-stimulated dHL-60, to a similar degree when treated with *miR-223* AS ODN, indicating *miR-223* expression is affected by PGN even though *miR-223* was knocked down by *miR-223* AS ODN (Fig 8B and C).

Furthermore, even when *miR-223* expression is strongly suppressed by *miR-223* AS ODN, if PGN stimulation (*S. aureus* recognition) is received, positive feedback occurs; thus, through increased IL-6 production, infection control becomes more effective.

*Il6* mRNA expression was markedly increased in non-PGN (4.9-fold increase) and PGN-stimulated (2.7-fold increase) *miR-223* AS ODN-treated dHL-60 cells compared with controls (Fig 8D). The concentration of IL-6 protein was significantly increased in conditioned media from *miR-223* AS ODN-treated dHL-60 cells compared with controls at each time point and this corresponded to *Il6* mRNA levels (Fig 8E).

Finally, to investigate how *miR-223* expression is regulated in human neutrophils infected with *S. aureus*, we investigated expression of transcription factors C/EBPα (Fazi *et al*, 2005; Fukao *et al*, 2007), RUNX1 (Fazi *et al*, 2007), and PU.1 (Fukao *et al*, 2007) that regulate *miR-223* expression using qPCR. The expression of *CEBPA*, but not *RUNX1* and *PU.1*, in dHL-60 was significantly decreased after 6-h stimulation with PGN compared with non-stimulated dHL-60 cells (Fig EV5A). Finally, we confirmed the physical interaction between C/EBPα and the *miR-223* promotor locus in dHL-60 cells after PGN stimulation because of the decreased levels of C/EBPα-*miR-223* promotor locus interaction compared with non-stimulated dHL-60 cells using a chromatin immunoprecipitation (ChIP)-qPCR assay (Fig 8F and G; Fig EV5B and C).

Non-activated, mature neutrophils, such as those in the circulation, highly express *miR-223* (Fig 8H), which is induced by C/EBPα and leads to the inhibition of *Il6* translation. By contrast, activated neutrophils exposed to *S. aureus* downregulate *miR-223* expression concomitant with decreased C/EBPα and the subsequent induction of *Il6* translation.

## Discussion

In the present study, we established novel Ago2-miRNAs purification methods using NGS from skin wound tissues and identified skin wound-induced miRNAs that might bind to target key wound mRNAs. We identified four miRNA candidates that appear to be wound inflammation-related by comparison of WT with $PU.1^{-/-}$ mouse wounds. Of these, we demonstrated that $miR-223^{Y/-}$ mice exhibited delayed sterile skin wound healing but that in *S. aureus*-infected wounds, the addition of $miR-223^{Y/-}$ neutrophils, or knockdown of *miR-223* with AS ODNs significantly improved healing. The

---

**Figure 8.    Human neutrophil IL-6 secretion is regulated by the C/EBPα-*miR-223* signaling pathway stimulated with *Staphylococcus aureus* PGN.**

A    qPCR analysis was used to determine the expression levels of *miR-223* in dHL-60 ($n = 3$).

B    *miR-223* AS ODN decreased *miR-223* expression in dHL60 (control; $n = 5$, control ODN and *miR-223* AS ODN; $n = 6$).

C    Downregulation of *miR-223* expression in dHL60 cells at 6 h after stimulation with PGN ($n = 6$).

D    qPCR analysis indicates the expression levels of *Il6* relative to beta-2-microglobulin (*B2m*) in non-stimulated and PGN-stimulated (6 h) dHL60 cells were markedly increased compared with controls ($n = 3$) after *miR-223* knockdown.

E    ELISA revealed non-stimulated and PGN-stimulated dHL60 cells constitutively secreted IL-6 ($n = 3–6$).

F    Decreased C/EBPα binding activity to the *miR-223* promotor after PGN stimulation (PGN) in dHL-60 cells compared with non-stimulated dHL-60 cells (C, control) revealed by ChIP assay. Full scans for electropherogram in Fig EV5C.

G    Quantification of anti-C/EBPα Ab-ChIP relative to non-stimulated dHL-60 cells using ChIP-qPCR ($n = 3$).

H    A model summarizing the interplay between *S. aureus*, C/EBPα, *miR-223*, and IL-6 secretion in neutrophils at wound sites.

Data information: All values represent the mean $\pm$ SD. Ordinary one-way ANOVA followed by Holm-Sidak multiple comparisons test was used to generate *P*-values (day 0 versus each day) (A). Unpaired *t*-tests (C–E, and G) followed by Welch's correction (B) were used to generate the *P*-values indicated. $*P < 0.05$, $**P < 0.01$, $****P < 0.0001$.

Source data are available online for this figure.

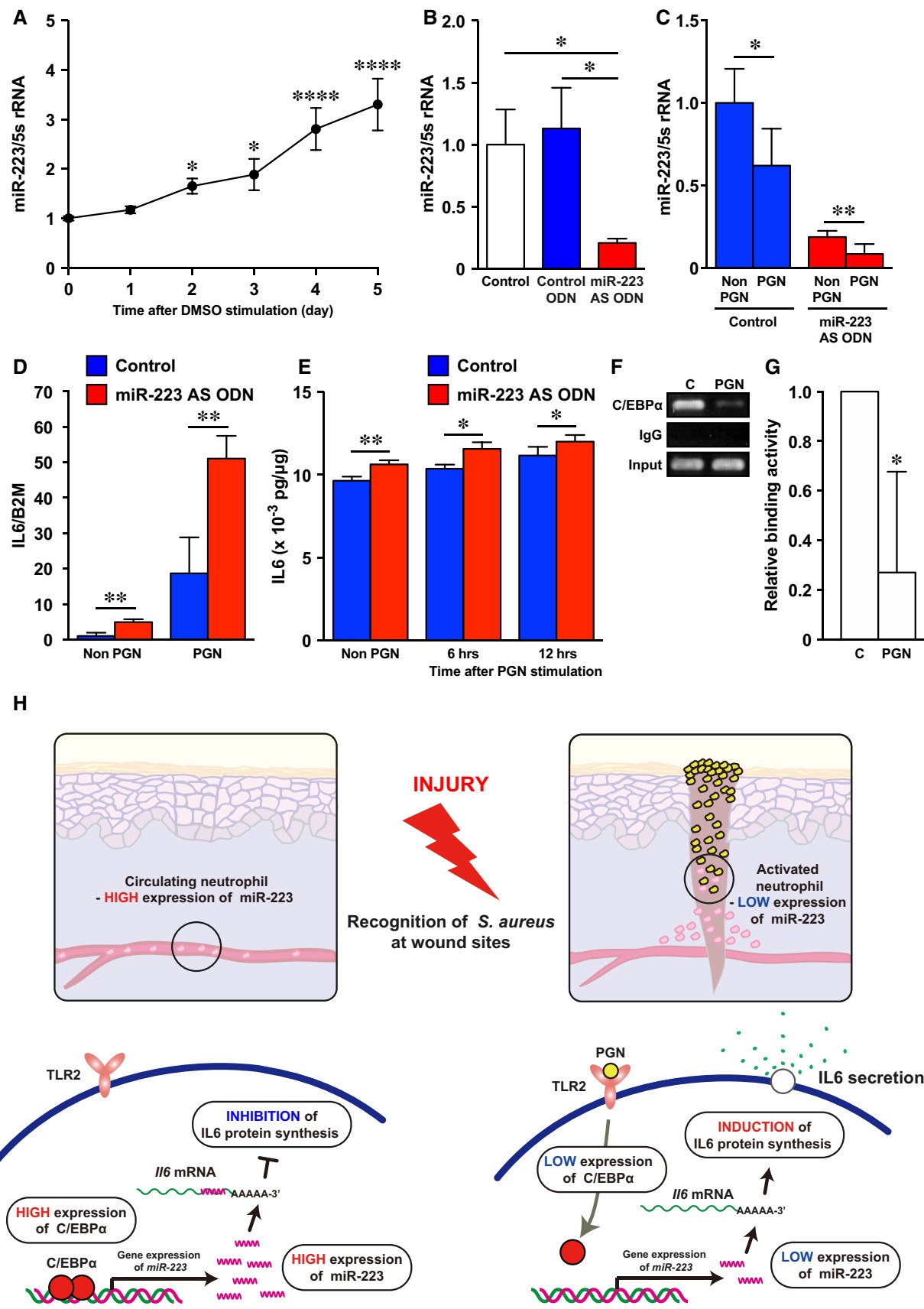

**Figure 8.**

*miR-223/Il6* pathway is regulated by C/EBPα and might have a critical role in *S. aureus* clearance. The present data provide a novel molecular insight into the function of *miR-223* in skin wound healing and suggest the potential therapeutic targeting of this miR in conditions such as infected skin ulcers.

Previously, over 30 inflammation and immune response-related miRNAs have been identified including *miR-142-3p*, *miR-142-5p*, and *miR-223* and our wound studies support a role for these miRNAs in aspects of the wound inflammatory response (O'Connell *et al*, 2012; Marques-Rocha *et al*, 2015). Recently, *miR-139-5p$^{-/-}$* mice were shown to develop colitis-associated tumorigenesis (Mao *et al*, 2015) and thus represent a potential clinical biomarker for cancer (Zhang *et al*, 2015). Our findings suggest that *miR-139-5p* is also associated with inflammatory responses and tissue repair.

Previous reports have shown that *miR-142* family members and *miR-223* may play critical roles in *in vivo* immune homeostasis. *miR-223* regulates granulocyte development and function (Johnnidis *et al*, 2008), and using *miR-142$^{-/-}$* mice, several groups revealed that *miR-142* regulates the hematopoietic development and function of mast cells, megakaryocytes, and lymphocytes (Mildner *et al*, 2013; Chapnik *et al*, 2014; Yamada *et al*, 2014; Sun *et al*, 2015). We further investigated the function of *miR-142* in skin wound healing by analyzing a newly-developed *miR-142* knockout mouse and showed that this miR regulates neutrophil migration at skin wound sites via the small GTPase regulation of its actin cytoskeleton (Tanaka *et al*, 2017).

In the current study, we identified a role for *miR-223* in the inflammatory phase of murine skin wound healing. Furthermore, we report the *miR-223* seed sequence is critical for the regulation of IL-6 expression and secretion. Recent studies have indicated roles for *miR-223* in numerous inflammation-related diseases, such as tuberculosis, inflammatory bowel disease, diabetes, and cancers (Chuang *et al*, 2015; Liu *et al*, 2015). Human and mouse studies by Fukao *et al* (2007) suggest *miR-223* might fine-tune the expression of numerous genes during inflammatory episodes. A recent study reported that *miR-223* does not bind to *Il6* mutation sites, suggesting *miR-223* directly targets *Il6* in myeloid cells (Dorhoi *et al*, 2013). Over twenty *miR-223* targets associated with immunity have been validated in humans and mice (Haneklaus *et al*, 2013). In addition, the downregulation of *miR-223* promoted *Il6* and *Il1b* transcription in a RAW264.7 macrophage cell line (Chen *et al*, 2012). Thus, *miR-223* might participate in trafficking pathways including the synthesis, storage, and secretion of bioactive substances and *de novo* synthesis of proinflammatory cytokines through transcriptional regulation (Sheshachalam *et al*, 2014).

It was reported that *IL6$^{-/-}$* mice exhibited delayed aseptic skin wound healing (Gallucci *et al*, 2000; Lin *et al*, 2003) and recombinant murine IL-6 improved skin wound healing in glucocorticoid-treated mice (Gallucci *et al*, 2001). In contrast, excess IL-6 causes inflammatory diseases; therefore, IL-6 receptor antibody (Tocilizumab) has been used as therapeutic agent against Castleman disease and rheumatic diseases in the clinic (Rubbert-Roth *et al*, 2018; Yoshizaki *et al*, 2018). Taken together, it appears that precise regulation of IL-6 levels is necessary at inflamed sites to regulate inflammation and tissue repair.

It is well established that neutrophils kill microbes through various strategies including the release of ROS generated by NADPH-dependent oxidase (Ellson *et al*, 2006). Previous studies reported that *miR-223$^{Y/-}$* neutrophils exhibited increased H$_2$O$_2$ production and were efficient at *Candida albicans* killing *in vitro* (Johnnidis *et al*, 2008). *miR-223* expression was markedly increased in the inflamed pelvic sites of individuals suffering from chlamydial genital infection (Yeruva *et al*, 2014), indicating *miR-223*-expressing cells might be involved in many infectious diseases. We wondered whether *miR-223*-expressing cells might play a role in *S. aureus*-infected wounds. Indeed, we found that *S. aureus*-infected *miR-223$^{Y/-}$* murine wounds healed faster than in WT mice during the early inflammation phase. Moreover, *Il6* and *Il1b* expressions, which contribute to *S. aureus* clearance in skin wound sites (Hruz *et al*, 2009; Cho *et al*, 2012) and in keratitis (Hume *et al*, 2006), were significantly higher in *miR-223$^{Y/-}$* wounds. We mirrored the efficacy of cell transplantation therapy against *S. aureus*-infected skin wounds using purified *miR-223$^{Y/-}$* neutrophils, by delivery of *miR-223* AS ODN knockdown of the same miRNA which similarly resulted in a better and faster healing of infected wounds in mice and has more immediate therapeutic applicability. Re-epithelialization and granulation tissue formation at the aseptic wound sites of *miR-223$^{Y/-}$* mice were markedly delayed in contrast to the *S. aureus*-infected skin wound sites of *miR-223$^{Y/-}$*, *miR-223$^{Y/-}$* neutrophil-transplanted, and *miR-223* AS ODN-treated groups, which were significantly improved compared with each control. This is likely a consequence of increased acute inflammatory responses (secondary effect), because *miR-223* was not expressed in wound-infiltrated fibroblasts or keratinocytes.

In this study, we found that activating human neutrophils by exposure to *S. aureus* peptide downregulated *miR-223* associated with the attenuation of C/EBPα that is required for *S. aureus* clearance *in vivo*. It has previously been reported that murine neutrophils produced IL-6 after PGN stimulation (Strassheim *et al*, 2005). C/EBPα expression was downregulated by PGN and lipopolysaccharide in microglial cells, whereas *Il6* and *Il1b* did not affect C/EBPα expression (Ejarque-Ortiz *et al*, 2007). Indeed, the TLRs-C/EBPα-*miR-223* signaling pathway is likely to be the predominant early phase signaling pathway driving bacteria clearance, rather than cytokine signaling.

Because we showed that *miR-223* is also upregulated in human inflamed tissues, it is tempting to speculate that these therapeutic approaches may also translate to human chronic wounds. Chronic wounds affect 6.5 million patients annually, and 25 billion USD is spent in the United States to combat this (Sen *et al*, 2009). Infected chronic wounds frequently show a poor response to treatment and are difficult to cure; therefore, there is an enormous economic and social impact worldwide. We hope that a deeper understanding of *miR-223* molecular mechanisms may aid the development of novel drugs and cell-based therapies to enhance the healing of chronically infected wounds.

# Materials and Methods

### Mice and wound model

All experiments were conducted according to the provisions of the Ethics Review Committee for Animal Experimentation at Nagasaki University. Mice were maintained in a barrier facility (temperature;

22–25°C, 12-h light/dark cycle) under specific pathogen-free conditions. Mice were fed *ad libitum* with Charles River-LPF diet (360 kcal/100 g; 13% fat calories, 26% protein calories, and 61% carbohydrate calories [Oriental Yeast, Tokyo, Japan]). *miR-223*$^{Y/-}$ mice were generated as described previously (Johnnidis *et al*, 2008). The *miR-223* locus is located on the X chromosome and is transcribed independently of any known genes, so that *miR-223*$^{Y/-}$ hemizygous male mice are completely deficient in mature *miR-223* expression. WT male mice (6–12 weeks) and *miR-223*$^{Y/-}$ male mice (B6.Cg-*Ptprc*$^a$ *Mir223*$^{tm1Fcam}$/J, 6–12 weeks [The Jackson Laboratory, Bar Harbor, ME, USA]) were anaesthetized and 2 or 4 full-thickness excisional wounds (4-mm biopsy punch; Kai Industries, Gifu, Japan) were aseptically made to the shaved dorsal skin. Generation of *PU.1*$^{-/-}$ mice was described previously (McKercher *et al*, 1996). One-day-old pups received anesthetic and full-thickness 1-cm incisional wounds were performed on the dorsal skin; then, the wounds were harvested.

Faust *et al* (2000) generated *lys*-EGFP mice as described previously. To generate male *lys*-EGFP-expressing *miR-223*$^{Y/-}$ mice, male *miR-223*$^{Y/-}$ mice were crossed with female *lys*-EGFP mice to produce lys-EGFP heterozygous-expressing male *miR-223*$^{Y/-}$ and female *miR-223*$^{+/-}$ mice. A second cross generated male *lys*-EGFP homozygous-expressing *miR-223*$^{Y/-}$ mice. Mice genotypes were defined by PCR as previously described (Faust *et al*, 2000; Johnnidis *et al*, 2008).

*Staphylococcus aureus* type strain (NBRC 100910) was obtained from the National Institute of Technology and Evaluation (Tokyo, Japan). In the *S. aureus*-infected group, mice were locally inoculated with *S. aureus* ($1 \times 10^8$ CFU per 10 μl in saline) at pre-skin wound sites followed by making the wound.

For wound area analysis, digital images of wound areas were measured by Photoshop CC (Adobe Systems, San Jose, CA, USA) and the mean wound area was calculated from two or four wounds from a single mouse (Mori *et al*, 2008, 2014; Tanaka *et al*, 2017).

### Extraction and purification for miRNA from skin wounds

Wounds in mice were harvested with a 6-mm biopsy punch, and miRNAs were extracted/purified using the microRNA Isolation Kit, Mouse Ago2 (Wako Pure Chemical Industries Ltd, Osaka, Japan) according to the manufacturer's instructions. Briefly, 4 wound tissues were placed in cell lysis buffer and homogenized for 5 min, incubated on ice for 15 min, and centrifuged at 14,000 *g* for 15 min at 4°C. Debris in supernatants was eliminated using an Ultrafree-MC 0.45-μm filter (Merck Millipore, Darmstadt, Germany). Filtrated supernatants were immunoprecipitated with anti-mouse Ago2 Ab for 3 h at 4°C. Samples were washed with cell lysis buffer 3 times, and miRNAs were eluted by elusion solution. miRNAs were purified and precipitated by standard phenol/chloroform extraction and ethanol precipitation methods.

### MiRNA library construction and high-throughput whole-miRNAs sequencing

Cloning of miRNAs was performed using a small RNA Cloning Kit (Takara Bio, Shiga, Japan) according to the manufacturer's instructions (Appendix Fig S1). Briefly, purified-miRNAs were treated with Bacterial Alkaline Phosphatase to dephosphorylate the 3′ end of

miRNAs at 37°C for 1 h, then purified and precipitated by standard phenol/chloroform extraction and ethanol precipitation methods. Dephosphorylated-3′ ends of miRNAs were bound to biotinylated-3′ adaptors using T4 RNA ligase at 15°C for 1 h, reacted with MAGNOTEX-SA (Takara Bio), a streptavidin conjugated-magnetic bead and then washed. Magnet bead-conjugated miRNAs were phosphorylated at the miRNA 5′ end using T4 polynucleotide kinase at 37°C for 30 min, bound with 5′ adaptors using T4 RNA ligase at 15°C for 1 h, and then washed. Finally, miRNAs were transcribed to cDNA and amplified using ExTaq Hot Start Version (Takara Bio). Specific primers were as follows: (Forward) 5′-AAAGATCCTG CAGGTGCGTCA-3′ and (Reverse) 5′-GTCTCTAGCCTGCAGGATC GATG-3′. Amplification was performed for 15 cycles with annealing at 60°C for 30 s, extension at 72°C for 30 s, and a final extension for 3 min. First PCR products were amplified with the same conditions as for the 1st PCR methods. Second PCR products (approximately 65 bp) were separated by electrophoresis on a 10% TBE gel stained with SYBR Gold nucleic acid gel stain (Invitrogen, Carlsbad, CA, USA). Second PCR products were extracted from the gel and purified using a small RNA gel Extraction kit (Takara Bio) according to the manufacturer's instructions. Purified 2$^{nd}$ PCR products were amplified using ExTaq Hot Start Version (Takara Bio) with the same conditions as for the 1$^{st}$ PCR methods. Specific primers were as follows: (Forward) 5′-AATGATACGGCGACCACCGAGATCTACAC TCTTTCCCTACACGACGCTCTTCCGATCTAAAGATCCTGCAGGTGC GTCA-3′ and (Reverse) 5′-CAAGCAGAAGACGGCATACGAGCTCTT CCGATCTGTCTCTAGCCTGCAGGATCGATG-3′.

Third PCR products (approximately 156 bp) were separated by electrophoresis on a 10% TBE gel stained with SYBR Gold nucleic acid gel stain (Invitrogen). Target PCR products were extracted from gels and purified using a small RNA gel Extraction kit (Takara Bio) according to the manufacturer's instructions. Purified miRNA libraries were subjected to Solexa sequencing system (Illumina, San Diego, CA, USA) according to the manufacturer's instructions. miRNA-Seq data reported are available in the DNA Data Bank of Japan (DDBJ, http://trace.ddbj.nig.ac.jp/DRASearch/) under accession no. DRA004094.

### Analysis of miRNA sequencing data

All individual sequence reads with base quality scores were trimmed and eliminated from ineffective sequencing in the initial data. Resulting sets of unique sequence reads were mapped onto the mouse genome with reference to Langmead *et al* (2009) using the UCSC Genome Bioinformatics Site. To identify sequence tags originating from known ncRNA (miRNA, other species miRNA, piRNA, rRNA, tRNA, snRNA, snoRNA, scRNA, miscRNA) and transcripts, we used miRBase (http://www.mirbase.org/index.shtml), NCBI Entrez Nucleotide database (http://www.ncbi.nlm.nih.gov/sites/en trez/query.fcgi?db=Nucleotide), and Reference Sequences (ftp:// ftp.ncbi.nih.gov/), as well as Ensembl Genome Browser (ftp://ftp. ensembl.org/pub/release55/).

### qPCR

Harvested tissues were homogenized by TissueLyzer II (Qiagen, Hilden, Germany), and total miRNAs and mRNAs were extracted/ purified using an miRNeasy Mini Kit (Qiagen) and RNeasy Plus

Universal Mini Kit (Qiagen), respectively, according to the manufacturers' instructions. Quantification of miRNAs was performed by TaqMan MicroRNA Assay (Applied Biosystems, Foster City, CA, USA) and the miRCURY LNA microRNA PCR system (Exiqon, Vedbaek, Denmark) using ABI PRISM 7900HT (Applied Biosystems). Quantification of mRNAs was performed with Thunderbird SYBR qPCR Mix (Toyobo Inc., Osaka, Japan) using ABI PRISM 7900HT (Applied Biosystems) (sequences of primers are shown in Appendix Table S6).

### Histology

Harvested tissues were fixed in 4% paraformaldehyde (PFA) overnight and embedded in paraffin. All specimens were cut to 4-μm-thick sections and subjected to hematoxylin and eosin (H&E), ISH, and IHC (Ab information in Appendix Table S5). IHC and quantification of macrophages and αSMA expression were performed as previously described (Mori *et al*, 2006, 2008, 2014). Observations were made via digital whole slide scanning system (Aperio AT Turbo; Leica Microsystems, Tokyo, Japan) confocal microscopy (C2+ system; Nikon Corp., Tokyo, Japan)]. Aperio eSlide Manager (Leica Microsystems), NIS-Elements C software version 4.13 (Nikon Corp.), AR software version 4.0 (Nikon Corp.), or IMARIS 7.7.2 (Bitplane, Zurich, Switzerland) were used for data analysis.

### ISH

*In situ* hybridization was performed using microRNA ISH buffer set and miRCURY LNA Detection 5′- and 3′-DIG-labeled probes (Exiqon) according to the manufacturer's instructions. In brief, 4% PFA perfusion-fixed tissues were embedded in paraffin. Six-μm sections were deparaffinized and incubated with Proteinase K solution (Dako, Glostrup, Denmark) for 10 min at 37°C. After washing in PBS, sections were dehydrated. Hybridization was performed using 40 nM of miRNA probe in microRNA ISH buffer (Exiqon) at 55°C for 2 h. Sections were rinsed in 5× SSC at 55°C for 5 min, twice with 1× SSC at 55°C for 5 min, twice with 0.2× SSC at 55°C for 5 min, and with 0.2× SSC at room temperature for 5 min. Sections were treated with blocking solution (Nacalai Tesque Inc., Kyoto, Japan) for 1 h at room temperature and then were incubated with anti-DIG Ab (1:800; Roche Diagnostics GmbH, Mannheim, Germany) in blocking solution (Nacalai Tesque Inc.) overnight at 4°C. Sections were developed using NTB/BCIP (Roche Diagnostics GmbH) at 30°C.

### Isolation of neutrophils and macrophages from skin wounds and bone marrow

Neutrophils and macrophages were isolated with a MicroBead Kit (Miltenyi Biotech Inc., Bergisch Gladbach, Germany) according to the manufacturer's instructions. Cells were reacted with anti-Ly-6G Ab and anti-CD11b Ab to isolate neutrophils and macrophages, respectively (Tanaka *et al*, 2017).

### *In vivo* live imaging analysis of EGFP-expressing neutrophil recruitment and MPO activity at wound sites

Neutrophil recruitment at skin wound sites was visualized and measured as previously described (Kim *et al*, 2008; Tanaka *et al*, 2017). In brief, WT:*lys*-EGFP mice and *miR-223*$^{Y/-}$:*lys*-EGFP mice were wounded and fluorescence intensity was monitored with IVIS Lumina II System (PerkinElmer, Waltham, MA, USA). Fluorescence intensities were expressed as the mean radiant efficiency ((p/sec/cm$^2$/sr)/(uW/cm$^2$)).

To visualize and measure MPO activity *in vivo*, mice were intraperitoneally injected with XenoLight RediJect Inflammation Probe (5 μl/g body weight; PerkinElmer) on day 1 and 3 after injury, and luminescence images were acquired at 10 min after probe injection with the IVIS Lumina II System (PerkinElmer). Luminescence intensities were expressed as the mean radiance (photons/second/cm$^2$/steradian). Living Image software (Perkin-Elmer) was used to analyze the intensities of each wound.

### Measurement of MPO, MIP-1α, and IL-6 protein concentrations

Extraction of total protein was performed as previously described (Mori *et al*, 2014). Briefly, harvested tissues were homogenized by TissueLyzer II (Qiagen) and T-PER Reagent (Thermo Fisher Scientific, Waltham, MA, USA), consisting of proteinase and dephosphorylation inhibitor. Sample proteins were filtered using an Ultrafree-MC 0.45-μm filter (Millipore). Concentrations of MPO and MIP-1α were measured with an MPO mouse ELISA kit (Abcam, Cambridge, UK) and a Bio-Plex Pro mouse cytokine G1 23-Plex panel (Bio-Rad Laboratories, Inc. Hercules, CA, USA), respectively.

Murine neutrophils were isolated from bone marrow with a MicroBead Kit (Miltenyi Biotech Inc.) according to the manufacturer's instructions. Cells were cultured in RPMI 1640 at 37°C in 5% CO$_2$ for 5 h. Concentrations of mouse IL-6 in conditioned media were measured by Mouse IL-6 Quantikine ELISA kit (R&D Systems, Minneapolis, MN, USA) according to the manufacturer's instructions.

### Live imaging analysis of neutrophil ROS production *in vitro*

Measurement of hypochlorite production was detected using APF (Goryo Chemical Inc., Sapporo, Japan) according to the manufacturer's instructions. In brief, 1 drop of tail vein peripheral blood was added to 2 ml of live cell imaging solution (Gibco, Carlsbad, CA, USA), and stained with 10 μM of APF for 30 min at room temperature. APF-loaded neutrophils were stimulated with 1 μl PMA, and fluorescence images were acquired every 1 min for 60 min using a confocal laser scanning unit microscope (C2+ system, Nikon Corp.) equipped with Plan Apo VC20x (0.75 NA), and the images were processed using IMARIS software (Bitplane).

### RNA-Seq

Wound tissues harvested at day 1 were homogenized by Tissue-Lyzer II (Qiagen) followed by total RNA and polyA$^+$ RNA purification using an RNeasy Plus Universal Mini Kit (Qiagen) and TruSeq Stranded Total RNA Library Prep Kit (Illumina), respectively. mRNA libraries were constructed with a TruSeq Stranded mRNA LT Sample Prep Kit (Illumina) according to the manufacturer's instructions (TruSeq Stranded mRNA Sample Preparation Guide Rev.E). RNA-Seq was performed with HiSeq 2500 (Illumina). RNA-Seq data reported are available in the DNA Data Bank of Japan under accession no. DRA004092.

## Assay for miRNA binding to the 3′-UTR of mRNA

The 3′-UTRs of miR-223 targets were screened using Strand NGS software (Strand Genomics, San Francisco, CA, USA). Vectors were constructed with pmirGLO Dual-Luciferase miRNA Target Expression Vector (Promega Corp., Madison, WI, USA) according to the manufacturer's procedures and as previously described (Tanaka et al, 2017). In brief, primers consisting of the 3′-UTRs of predicted miR-223 family target sequences and appropriate restriction sites were synthesized, annealed, and cloned downstream of the firefly luciferase reporter (luc2) gene in pmirGLO. Sequences were as follows: Il6 sense (78–106) 5′-aaacCTTATGTTGTTCTCTACGAAGAA CTGACAt-3′; and Il6 antisense (106–78) 5′-ctagaTGTCAGTTCTTCG TAGAGAACAACATAAGgttt-3′. Capital and lowercase letters indicate the 3′-UTR and restriction sites (PmeI and XbaI), respectively.

Double-strand miR-223 mimic and miR-223 point mutation mimics were purchased from GeneDesign Inc. (Osaka, Japan). Sequences were as follows (underline indicates a mutation point): miR-223-3p 5′-UGUCAGUUUGUCAAAUACCCCA-3′; miR-223-3p point mutation (No. 1) 5′-UGCCAGUUUGUCAAAUACCCCA-3′; miR-223-3p point mutation (No. 2) 5′-UGCGAGUUUGUCAAAUACCCCA-3′; and miR-223-3p point mutation (No. 3) 5′-UGCGAGCCUGUC AAAUACCCCA-3′.

3T3 cells were co-transfected with a miR-223 mimic and reporter vector using Lipofectamine 3000 (Life Technologies). Luciferase activity was measured with a Dual-Glo Luciferase Assay System (Promega Corp.) using the manufacturer's procedure.

## Neutrophil transplantation

Neutrophils were isolated from bone marrow using a Neutrophil Isolation Kit (Miltenyi Biotech Inc.) according to the manufacturer's instructions. Isolated neutrophils ($1 \times 10^6$ cells/50 µl in saline) were locally applied at day 1 after inoculating S. aureus to wound sites.

## Generating miR-223 AS ODN

To obtain mature miR-223-expressing vector, vectors were constructed with pmirGLO Dual-Luciferase miRNA Target Expression Vector (Promega Corp) according to the manufacturer's procedures. In brief, oligos consisting of the miR-223 sequences and appropriate restriction sites were synthesized, annealed, and cloned downstream of the luc2 gene in pmirGLO. Sequences were as follows: miR-223 sense 5′-aaacTGTCAGTTTGTCAAATACCCCAt-3′; and miR-223 antisense 5′-ctagaTGGGGTATTTGACAAACTGACAgttt-3′. Capital and lowercase letters indicate the mature miR-223 and restriction sites (PmeI and XbaI), respectively.

Control ODN and miR-223 AS ODN were originally designed and had sequences as follows: Control ODN 5′-G(L)^5(L)^A(L)^G(L) ^a^t^t^g^t^a^c^g^a^5(L)^T(L)^5(L)^A(L)^5(L)-3′; and miR-223 AS ODN 5′-G(L)^G(L)^T(L)^A(L)^t^t^t^g^a^c^a^a^a^5(L)^T(L)^G(L) ^A(L)^5(L)-3′.

L, ^, and 5(L) indicate LNA, phosphorothioated ODN, and LNA-modified 5-methylcytosine, respectively (Fig 6A).

3T3 cells were co-transfected with control ODN or miR-223 AS ODN and the reporter vector using Lipofectamine 3000 (Life Technologies). Luciferase activity was measured with a Dual-Glo Luciferase Assay System (Promega Corp.) using the manufacturer's procedure.

## miR-223 knockdown at skin wound sites using miR-223 AS ODN

For in vivo experiments involving ODN delivery, ODNs [10 µM in 50 µl poloxamer P407/P188 (25%/4%; BASF, Ludwigshafen am Rhein, Germany) binary thermosensitive hydrogel and 30% Pluronic F-127 gel (Sigma-Aldrich, St. Louis, MO, USA), which acts as a slow release vehicle (Mori et al, 2006, 2008, 2014)] were topically applied at 1 day after inoculation with S. aureus.

## Human skin sample

Human skin samples were harvested from Japanese patients at the time of surgery, and diagnosis was confirmed by routine pathological examination (Appendix Table S4). All experiments were conducted with the approval of the ethics committee of Nagasaki University Graduate School of Biomedical Sciences, in accordance with the WMA Declaration of Helsinki and the Department of Health and Human Services Belmont Report.

## Induction of neutrophilic differentiation and analysis for gene expression stimulation with S. aureus PGN

The HL-60 cell line (RBRC-RCB0041) was provided by RIKEN BRC through the National Bio-Resource Project of MEXT, Japan. HL-60 cell culture and neutrophilic differentiation were previously described (Shuto et al, 2007). Neutrophilic differentiation was induced by exposing HL-60 cells to 1.3% DMSO for 5 day. At 3 day after exposure to 1.3% DMSO, HL-60 cells were transfected with control or miR-223 AS ODN using Lipofectamine 3000 (Life Technologies) in Opti-MEM with 1.3% DMSO (Life Technologies). On 2 day after transfection, cells and conditioned media were harvested and applied to gene expression assays and ELISA (Human IL-6 ELISA Kit, Sigma-Aldrich, St. Louis, MO, USA).

## ChIP-qPCR assay

ChIP-qPCR assay was performed with ChIP-IT PBMC (Active Motif, Carlsbad, CA, USA) according to the manufacturer's instructions. Briefly, crosslinking of proteins to DNA was achieved by direct treatment of 1% formaldehyde to non-stimulated or PGN-stimulated dHL60 cells ($1 \times 10^7$ cells) for 15 min at room temperature. Chromatin was sonicated with a Covaris M220 (Covaris Inc., Woburn, MA, USA). Sonication conditions were as follows: duty factor 5%, peak incident power 75, cycles per burst 200, water temperature 7°C, and duration 15 min. Sonicated chromatin was immunoprecipitated for 16 h at 4°C with anti-C/EBPα Ab (GeneTex Inc., Irvine, CA, USA), anti-Histone H3 Ab (Cell Signaling Technology Inc., Danvers, MA, USA), and anti-IgG isotype control Ab (Appendix Table S5).

ChIP-PCR was performed with GoTaq Master Mix and Polymerase (Promega Corp., Madison, WI, USA) according to the manufacturer's instructions. In brief, ChIP-DNA fragments (0.25 µg/reaction) were amplified by PCR using the following primers that

**The paper explained**

**Problem**
A neutrophil influx is necessary for effective wound repair and essential for fighting off wound infection. However, an excessive inflammatory response is bad for healing. It is important to identify the regulators of wound inflammation that determine the extent and level of neutrophil activation to therapeutically target this response in sterile and infected wound scenarios.

**Results**
We identified a wound inflammation-related miR, *miR-223*, that appears to be a master regulator of neutrophil homeostasis and dampens activation in sterile wounds to prevent chronic inflammation. *miR-223* knockout (*miR-223$^{Y/-}$*) mice exhibited impaired healing of chronic wounds, but markedly faster healing of *Staphylococcus aureus*-infected wounds compared with infected wild-type (WT) mice; moreover, cell transplantation therapy using *miR-223$^{Y/-}$*-derived neutrophils and *miR-223* antisense oligodeoxynucleotide knockdown in wounds markedly improved the healing of infected WT wounds.

**Impact**
Our findings suggest that targeting *miR-223* might be of therapeutic benefit to enhance the healing of infected wounds in the clinic.

recognize the human C/EBPα binding miR-223 promotor site as described (Fazi *et al*, 2005; Appendix Table S6): ChIP-PCR primer (Forward) 5′-GCCCTCTTTGTTGATGTGTC-3′ and primer#1 (Reverse) 5′-GGCAGCTATTAAAGTGCCCT-3′ for the −1,030/−810 fragment with respect to the end of the pre-*miR-223* sequence (210 bp). Annealing was performed at 55°C followed by 35 amplification cycles. PCR products were separated by electrophoresis on 2% agarose gels (Fig 8F; Fig EV5D). ChIP-qPCR was performed with Thunderbird SYBR qPCR Mix (Toyobo Inc.) using an ABI PRISM 7900HT Sequence Detection System.

**Statistical analysis**

Data are shown as the means ± SD or SEM (Fig 4G only). The statistical significance of differences between means was assessed by ANOVA, followed by Tukey's test for multiple comparisons, Dunnett's multiple comparisons test, the Mann–Whitney *U*-test or by an unpaired Student's *t*-test followed by Welch's test when only two groups were analyzed. Multiple comparison tests were performed using two-way ANOVA followed by Sidak multiple comparisons test or by Holm-Sidak multiple comparisons test using GraphPad Prism software (GraphPad Software, San Diego, CA, USA) (Appendix Table S7).

# Data availability

miRNA-Seq data reported are available in the DNA Data Bank of Japan (DDBJ) under accession no. DRA004094 (http://ddbj.nig.ac.jp/DRASearch/submission?acc = DRA004094). RNA-Seq data reported are available in the DDBJ under accession no. DRA004092 (http://ddbj.nig.ac.jp/DRASearch/submission?acc = DRA004092).

**Expanded View** for this article is available online.

# Acknowledgements

We thank Drs. Thomas Graf (Gene Regulation, Stem Cells and Cancer Program, Centre for Genomic Regulation, Barcelona, Spain), Shintaro Hashimoto, and Masaki Honda (Kumamoto University, Japan) for providing *lys*-EFGP mice. We appreciate comments from Dr. Eun Seong Hwang (University of Seoul, Korea) regarding the experiments. We thank Ms. Utako Kikutani for modifying the figure in Synopsis. This work was supported in part by the Japan Society for the Promotion of Science (Grant-in-Aid for Research Activity Start-up, 20890258; Grants-in-Aid for Young Scientists (A), 21689049 and 24689069; Challenging Exploratory Research, 23650484 and 25560055 to R. M. and 26670773 to H. Y.; Grant-in-Aid for Scientific Research (B), 16H05493 to R. M.; Grants-in-Aid for Encouragement of Scientists, 15H00601 to M. d. K.; Grants-in-Aid for Young Scientists (B), 26861503 and 16K20361 to K. T., by The Takeda Science Foundation (R. M.), The Uehara Memorial Foundation (R. M.), The Nakatomi Foundation (R. M.), The Wellcome Trust (Senior Investigator Award 097791/Z/11/Z to P. M.), and The Royal Society (International Joint Project, R. M. and P. M.)).

# Author contributions

MK performed experiments using *miR-223$^{Y/-}$* mice and wrote part of the manuscript. KT, TU, MO, SN, and ST performed qPCR and ISH analysis. SN performed ChIP assay. YM and YO performed experiments using *PU.1$^{-/-}$* mice. KT, SK, TN, and YS performed NGS. TK and SP managed the breeding of mice. KT and HY contributed to the generation of *miR-223$^{-/-}$:lys*-EGFP mice. KIk performed immunoblotting and ELISA analysis. RT, YK, and TH developed the PB gel. HT designed *miR-223* AS ODNs. HH and KIw prepared human samples. KT, PM, and IS advised on all experiments and edited the manuscript. RM designed the research, performed experiments, wrote, and edited the manuscript.

# Conflict of interest

The authors declare that they have no conflict of interest.

# For more information

*miR-139* in Mouse Genome Informatics (MGI): http://www.informatics.jax.org/marker/MGI:2676824; *miR-142* in MGI: http://www.informatics.jax.org/marker/MGI:2676827; and *miR-223* in MGI: http://www.informatics.jax.org/marker/MGI:2684360.

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
