## [Review Process File · EMBO Molecular Medicine]

Targeting miR-223 in neutrophils enhances the clearance of *Staphylococcus aureus* in infected wounds

Maiko de Kerckhove, Katsuya Tanaka, Takahiro Umehara, Momoko Okamoto, Sotaro Kanematsu, Hiroko Hayashi, Hiroki Yano, Soushi Nishiura, Shiho Tooyama, Yutaka Matsubayashi, Toshimitsu Komatsu, Seongjoon Park, Yuka Okada, Rina Takahashi, Yayoi Kawano, Takehisa Hanawa, Keisuke Iwasaki, Tadashige Nozaki, Hidetaka Torigoe, Kazuya Ikematsu, Yutaka Suzuki, Katsumi Tanaka, Paul Martin, Isao Shimokawa, and Ryoichi Mori

Review timeline:

Submission date:	22 February 2018
Editorial Decision:	18 April 2018
Revision received:	13 July 2018
Editorial Decision:	18 July 2018
Revision received:	28 July 2018
Accept:	7 August 2018

Editor: Lise Roth

Transaction Report:

1st Editorial Decision

18 April 2018

Thank you for the submission of your manuscript to EMBO Molecular Medicine. We have now heard back from the two referees whom we asked to evaluate your manuscript.

As you will see from the reports below, both referees are positive and support publication of the article in EMBO Molecular Medicine pending appropriate revisions. Addressing the reviewers concerns in full will be necessary for further considering the manuscript in our journal. Particular attention should be given to spelling and grammatical errors. EMBO Molecular Medicine encourages a single round of revision only and therefore, acceptance or rejection of the manuscript will depend on the completeness of your responses included in the next, final version of the manuscript.

Please also contact us as soon as possible if similar work is published elsewhere. If other work is published, we may not be able to extend the revision period beyond three months.

I look forward to receiving your revised manuscript.

***** Reviewer's comments *****

Referee #1 (Remarks for Author):

In this paper the authors show, by various approaches, that miR-223 has an active role in wound repair, in particular in *Staphylococcus aureus* infected wounds. Using both a knock-out model for miR-223 (miR-223Y^{-/-}) and antisense oligodeoxynucleotides (AS ODN), the authors are able to decrease the time of healing, either by treating the wounds with miR-223Y^{-/-}-derived neutrophils or by administering a gel containing AS ODN. They also show that miR-223 directly binds to IL6 and that the expression of miR-223 is regulated by C/EBP α . Overall, the authors demonstrate that miR-223 could be seen as a potential therapeutic target, especially in the case severe chronic *S. aureus*-infected skin wounds.

This is a comprehensive and straightforward paper with a considerable amount of work. The experiments and the strategy to decipher the role of miR-223 in healing of infected wounds are appropriate and well designed. Of note, this paper is technically of outstanding quality, in particular regarding the purification system to isolate miRNAs and the approach to design the AS ODN. Most of the experiments support the conclusions and the potential therapeutic possibilities. The scheme included in the last figure is well appreciated and helpful for the overall interpretation of the data. Nevertheless, some issues should be addressed to strengthen the paper.

What was the rationale for choosing the subset of miRs displayed in figure 1A and B? What about the other identified top-candidates?

Information about the antibody to detect neutrophils by IHC should be given (Figure 2).

Although the authors have estimated the re-epithelialization in miR223y^{-/-} mice, information about the granulation tissue might be provided.

In Figure 2E and 5A, the selected images of wound closure are not representative of the results displayed in the related graph.

The paragraph regarding the regulation of acute inflammatory responses at wound sites deserves clarifications. Indeed, the experiments showing *in vivo* imaging of EGFP-expressing neutrophils (Fig. 3A and B) in WT and miR223y^{-/-} mice are over interpreted. The differences between the 2 conditions are minimal, probably resulting from a low number of animals. Based on these data, the conclusion "both delayed onset but subsequent impaired resolution of the acute wound inflammatory responses in miR223y^{-/-} mice" might be revised. Moreover, it does not fit with the MPO experiments showing a peak of neutrophil activity at day 1 (Fig. 3C-E) whereas the amount of neutrophils is maximal at day 3 (Fig. 3A and B).

The experiment investigating the role of miR-223 on IL6 expression at wound site cannot be ascribed exclusively to neutrophils since the analyses are performed on the entire wound site (Fig. 4). A representative picture of IL6 immunostaining in the miR223y^{-/-} mice (Fig. 4A) should be displayed.

In Figure 6 and the related methods, the sequence of the LNA-modified AS ODN does not match the aligned miR-223 AS ODN sequence below. Please check.

The authors should explain the relevance of using both PGN and miR-223 AS ODN (Fig. 7C).

Abbreviations such as mmu and hsa should be defined.

The paper should be revised for typing and grammatical errors, in the text and figures (e.g. figure 4G).

Referee #2 (Comments on Novelty/Model System for Author):

De Kerckhove et al. identified Ago-2-bound miRNAs in mouse skin wounds, which included - among others - miR-223. This miRNA was upregulated during the early inflammatory phase of wound healing and is expressed mainly by neutrophils. Functional studies revealed that miR-223Y^{-/-} mice have impaired healing of sterile wounds, but enhanced healing of *S. aureus* infected wounds, most likely due to stronger activation of neutrophils and enhanced production of IL-6 by these cells.

The potential therapeutic relevance of these results was demonstrated by knock-down of miR-223 at the wound site and by application of miR-223Y⁻ neutrophils, which resulted in enhanced healing of *S. aureus* infected wounds.

A role of miR-223 in inflammation and infection control had previously been demonstrated by others, and IL-6 had previously been identified as a miR-223 target. Therefore, these aspects are not completely novel. However, a role of miR-223 in wound healing has not been demonstrated. In particular, the role of this miRNA in healing of infected wounds is novel and interesting and of potential medical importance. However, there are also a few problems with the manuscript, which are summarized below.

- 1.) The authors should provide more information on the miR-223Y⁻ mice in Materials and Methods. In particular, it should be mentioned that these hemizygous mice are completely deficient in miR-223 (at least according to the original publication).
- 2.) Since the manuscript focuses on miR-223, the authors should show expression of this miRNA during the whole time course of wound healing. There may be a second peak of miR-223 expression and this would be important for the interpretation of the wound healing data in the mutant mice.
- 3.) It should be clarified that the early increase in miR-223 in skin wounds results from the infiltration of neutrophils and is most likely not a result of a real upregulation in immune cells.
- 4.) Fig. 2A: The authors should mention which antibody they used for the detection of neutrophils - Ly6G? In addition, they should mention in the legend that the area indicated with a rectangle is shown at high magnification below.
- 5.) Fig. 2D: The information in this figure is limited, since there is no comparison with other cells at the wound site, in particular fibroblasts and keratinocytes. Given the delayed reepithelialization in the mutant mice, it is particularly important to determine if this is a cell autonomous effect that results from expression of miR-223 in keratinocytes or a secondary effect resulting from enhanced inflammation (more likely).
- 6.) Fig. 2D-H: The authors should confirm that miR-223 is indeed not expressed in the mutant mice. Wound healing is a combination of reepithelialization and wound contraction - is contraction also affected in the mutant mice?
- 7.) Fig. 3B is not convincing - there is only a statistically significant difference at the 3h time point and at the 3d time point - this needs to be formulated more carefully. The different functionality of the neutrophils may be more important than this minor difference in number. I am also not convinced that there is impaired resolution, since no difference was seen at day 7.
- 8.) Fig. S3A: Please show representative stainings.
- 9.) The authors should show IL-6 mRNA levels in non-stimulated and activated neutrophils of wt and miR-223 mutant mice - this would further support the regulation of IL-6 by miR-223. The qPCR shown in Fig. 4B only shows that IL-6 expression is enhanced in total wounds of miR-223 mutant mice, which may be secondary to the enhanced numbers of neutrophils (and not a real regulation by the miRNA).
- 10.) The upregulation of IL-6 is unlikely to explain the impaired healing in miR-223 mutant mice (IL-6 knock-out mice have impaired wound healing; Lin et al., 2003; IL-6 promotes wound healing in glucocorticoid-treated mice; Gallucci et al., 2001). Therefore, the mechanism underlying the impaired healing in the miR-223 mutant mice under sterile conditions remains unclear. This should at least be discussed.
- 11.) Fig. 5A and G: In addition to the macroscopic analysis, the authors should show H/E-stained sections from 7-day and 14-day wounds (and ideally use them to determine if there is an effect on reepithelialization and contraction). Given the rather high error bars, analysis of these histological parameters would clearly strengthen the data. At least one would like to get an idea about the histological features of the healing and healed wounds.
- 12.) Fig. 5G,H: Is it possible to determine how long the neutrophils used for treatment remain in the wound tissue? The cells could be labeled for this purpose.
- 13.) Fig. 6: The authors should verify that miR-223 is indeed downregulated by the ODNs at the wound site and check if IL-6 is upregulated.
- 14.) The paragraph describing the results shown in Fig.7 includes various errors in spelling and grammar and is therefore difficult to read. Most importantly, the results do not allow the conclusion that miR-223 is regulated by C/EBP α in response to *S. aureus*. To test this possibility, the authors would have to overexpress C/EBP α and determine if the PGN-mediated decline in miR-223 is rescued. Therefore, the scheme shown in Fig. 6G is not fully supported by the data.

Referee #2 (Remarks for Author):

This is an interesting manuscript and the data are generally convincing. However, additional experiments and some rewriting are required for publication in EMM.

1st Revision - authors' response

13 July 2018

Reviewer comments:

Reviewer: 1

Referee #1 (Remarks for Author):

Comment 1: *In this paper the authors show, by various approaches, that miR-223 has an active role in wound repair, in particular in Staphylococcus aureus infected wounds. Using both a knock-out model for miR-223 (miR-223Y/-) and antisense oligodeoxynucleotides (AS ODN), the authors are able to decrease the time of healing, either by treating the wounds with miR-223Y--derived neutrophils or by administering a gel containing AS ODN. They also show that miR-223 directly binds to IL6 and that the expression of miR-223 is regulated by C/EBP α . Overall, the authors demonstrate that miR-223 could be seen as a potential therapeutic target, especially in the case severe chronic S. aureus-infected skin wounds.*

This is a comprehensive and straightforward paper with a considerable amount of work. The experiments and the strategy to decipher the role of miR-223 in healing of infected wounds are appropriate and well designed. Of note, this paper is technically of outstanding quality, in particular regarding the purification system to isolate miRNAs and the approach to design the AS ODN. Most of the experiments support the conclusions and the potential therapeutic possibilities. The scheme included in the last figure is well appreciated and helpful for the overall interpretation of the data. Nevertheless, some issues should be addressed to strengthen the paper.

What was the rationale for choosing the subset of miRs displayed in figure 1A and B? What about the other identified top-candidates?

Response: First, we screened candidates for inflammatory-related miRNAs using the results from next generation sequencing (NGS) and found nine candidates for inflammation-related miRNAs that peaked on day (d) 1 after injury (Appendix Table S2.). Next, we rechecked NGS data using qPCR and confirmed the expression of the top 8 candidate miRs (*miR-147*, *miR-223*, *miR-129-3p*, *miR-139-5p*, *miR-21**, *miR-340-5p*, *miR-142-3p*, and *miR-142-5p*) was significantly increased compared with intact skin, indicating that these miRs might be candidates for inflammation-related genes. We could not confirm the expression of *miR-486* (fold change 4.51), suggesting that the cutoff value was >4.5 in our NGS results. We next tested for the expression of our 8 candidates in *PU.1*^{-/-} mice that lack an inflammatory response in skin wound sites versus WT sibs. This approach allowed us to definitively confirm that *miR-223*, *miR-142-3p*, *miR-142-5p*, and *miR-139-5p* are inflammation-related miRNAs in skin wound healing because these molecules were not expressed at wound sites in *PU.1*^{-/-} mice.

As suggested, we have summarized the other identified top candidates in Appendix Table S3. One of these, miR, *miR-21*, was more highly expressed during skin wound healing compared with the other miRs and this expression was markedly increased on d 3 (4.70), 7 (8.64), and 14 (5.77) compared with intact skin, suggesting *miR-21* might be involved in skin wound healing. Indeed, the function of *miR-21* in skin wound healing has been well studied (Han Z et al, *J Cell Biochem*, 2017, PMID: 28374893) (Pastar I et al, *J Biol Chem*, 2012, PMID: 22773832) (Wang T et al, *Am J Pathol*, 2012, PMID: 23159215) (Yang X et al, *Int J Biol Sci*, 2011, PMID: 21647251). We are currently investigating other candidate skin wound healing-related miRs using the NGS results.

Comment 2: *Information about the antibody to detect neutrophils by IHC should be given (Figure 2).*

Response: In accord with the comments by you and another reviewer, we have now added information for the neutrophil antibody to the Fig 3A legend (page 50, line 21) and Appendix Table S5.

Comment 3: *Although the authors have estimated the re-epithelialization in miR223y/- mice, information about the granulation tissue might be provided.*

Response: In accord with the comments by you and another reviewer, we investigated the area of granulation tissues at d 7 and 14 in aseptic wound sites (Fig EV1A-EV1C). We found that aseptic wound sites in *miR-223*^{-/-} mice were significantly increased compared with WT mice. The use of histological analysis allowed us a better understanding compared with gross appearance. The gross appearance of wound closure at d 14 in the wound sites of *miR-223*^{-/-} mice was not altered compared with WT mice. However, wound contraction might be related to the area of granulation tissue. To investigate wound contraction we performed immunohistochemistry (IHC) for α -smooth muscle actin (α SMA), a marker of contracting myofibroblasts, according to our previous report (Mori et al, J Cell Sci, 2006, PMID: 17158921). Expression of α SMA at aseptic wound sites in *miR-223*^{-/-} mice at d 7 were markedly decreased compared with WT mice. We have modified the text accordingly (page 9, line 12 to 20) and Fig EV1.

Comment 4: *In Figure 2E and 5A, the selected images of wound closure are not representative of the results displayed in the related graph.*

Response: As mentioned, we have modified Fig 3E (previously Fig 2E) and Fig 6A (previously Fig 5A).

Comment 5: *The paragraph regarding the regulation of acute inflammatory responses at wound sites deserves clarifications. Indeed, the experiments showing in vivo imaging of EGFP-expressing neutrophils (Fig. 3A and B) in WT and miR223y/- mice are over interpreted. The differences between the 2 conditions are minimal, probably resulting from a low number of animals. Based on these data, the conclusion "both delayed onset but subsequent impaired resolution of the acute wound inflammatory responses in miR223y/- mice" might be revised. Moreover, it does not fit with the MPO experiments showing a peak of neutrophil activity at day 1 (Fig. 3C-E) whereas the amount of neutrophils is maximal at day 3 (Fig. 3A and B).*

Response: We agree with your comments and have revised the paragraph regarding the onset and resolution of inflammatory responses in WT and *miR-223*^{-/-} mice according to your advice. In Fig 4A and 4B (previously Fig 3A and 3B), we wanted to show the neutrophil influx into the wound site over time using EGFP green fluorescent labeled neutrophils. Kim and colleagues (J Invest Dermatol, 2008), reported that neutrophil influx after skin wounding in *lys-EGFP* mice increased most rapidly over the initial 12 h and reached a maximum between d 1 and d 3. It then decreased precipitously at d 5 (Fig 2 and 3 in Kim et al., J Invest Dermatol, 2008). Our results are similar to theirs; at 12 h in WT mice, the neutrophil influx had started to increase and at d 3 the influx had peaked. In contrast, the rate of influx in *miR-223*^{-/-} mice appeared somewhat slower (3 h) although it also peaked at d 3. By d 3, the influx rate of *miR-223*^{-/-} mice had surpassed that of WT mice, and the neutrophil influx became excessive. We also modified the representative results of *in vivo* fluorescent images of EGFP-expressing neutrophils in skin wound sites as reflected in the related graph (Fig 4A).

Regarding MPO (Fig 4C-4E), we measured MPO to show the change in neutrophil function in *miR-223*^{-/-} mice *in vivo*. Because mature neutrophils do not produce new MPO, and only activated neutrophils activate MPO, we could assess neutrophil responses to wounding stimuli. Klebanoff (J Leukoc Biol. 2005), reported that MPO synthesis in neutrophil development starts in the promyelocyte stage and ends as enclosed azurophil granules at the beginning of the myelocyte stage; thus, mature neutrophils no longer produce MPO (Klebanoff SJ, J Leukoc Biol. 2005). MPO imaging using an inflammation probe measured the MPO activity produced by activated neutrophils; therefore, the peak of this image at d 1 shows the response to the wounding stimulus. At d 1, the wounding stimulus induces neutrophils to release MPO according to phagocyte function, such that MPO reaches a peak at d 1. At d 3, as the wounding stimulus decreases and MPO is no longer needed, the amount of MPO decreases. Fig 4E indicates that the amount of MPO measured by ELISA decreased relative to the time post-wounding stimulus.

We have also examined *in vitro* reactive oxygen species (ROS) production in neutrophils (Fig 4F and 4G). We performed live cell imaging analysis using confocal microscopy to dissect time-dependent neutrophil activation. ROS production in peripheral blood neutrophils (PBNs) derived from *miR-223*^{-/-} mice was slowly activated and interestingly, at 60 min *miR-223*^{-/-} PBNs exhibited increased ROS production compared with WT PBNs.

Collectively, our *in vivo* and *in vitro* analyses indicate that *miR-223* regulates the acute inflammatory response at wound sites and subsequently affects macrophage infiltration at wound sites. We have modified the text accordingly (page 10, line 1 to page 12, line 1).

Comment 6: *The experiment investigating the role of miR-223 on IL6 expression at wound site cannot be ascribed exclusively to neutrophils since the analyses are performed on the entire wound*

site (Fig. 4). A representative picture of IL6 immunostaining in the *miR223*^{-/-} mice (Fig. 4A) should be displayed.

Response: As you suggest IL-6 might be expressed by various skin wound-related cells such as neutrophils, epidermal keratinocytes, macrophages, Langerhans' cells, and fibroblasts (Paquet P et al, Int Arch Allergy Immunol, 1996, PMID: 8634514) (Sato Y et al, Int J Legal Med, 2000, PMID: 10876984) (Gallucci RM et al, FASEB J, 2000, PMID: 11099471). Our *PU.1*^{-/-} study suggests that *miR-223* is not expressed in keratinocytes and fibroblasts at wound sites during the early wound response; therefore, we speculated that keratinocytes and fibroblasts probably do not affect IL-6 expression related to the deletion of *miR-223*. Furthermore, keratinocytes and fibroblasts did not migrate to wound sites in the early acute inflammatory phase (d 1); for these reasons we focused on IL-6 expression in neutrophils. We have demonstrated that *miR-223* is predominantly expressed by neutrophils in the acute inflammatory phase using IHC and *in situ* hybridization (Fig 3A-3C), suggesting that it is largely neutrophils that exhibit altered IL-6 expression.

As you suggested, we have now performed IHC for IL-6 using double immunofluorescence staining of d 1 wound sites in *miR-223*^{+/+} and WT mice. We confirmed that the expression of IL-6 (red color) in wound-infiltrated neutrophils (green color) in *miR-223*^{+/+} mice were markedly increased compared with WT mice. We modified the text accordingly (page 12, line 11 to 13) and changed Fig 5A (previously Fig 4A).

Comment 7: In Figure 6 and the related methods, the sequence of the LNA-modified AS ODN does not match the aligned *miR-223* AS ODN sequence below. Please check.

Response: As mentioned, we modified the sequence of the LNA-modified AS ODN to match the aligned *miR-223* AS ODN (Fig 7A).

Comment 8: The authors should explain the relevance of using both PGN and *miR-223* AS ODN (Fig. 7C).

Response: We wanted to show that *miR-223* expression in *miR-223* AS ODN-treated dHL60 cells was significantly downregulated after PGN stimulation even though *miR-223* was knocked down similar to the control experiments using normal dHL60 cells. We now explain the relevance of using both PGN and *miR-223* AS ODN more carefully in the text (page 17, line 16 to 17).

Comment 9: Abbreviations such as *mmu* and *hsa* should be defined.

Response: Thank you, we now have defined *mmu* and *hsa* in the text (page 51, line 2 and 6).

Comment 10: The paper should be revised for typing and grammatical errors, in the text and figures (e.g. figure 4G).

Response: As suggested, typing and grammatical errors in the text and all Figures have now been checked by an English editing company.

Reviewer comments:

Reviewer: 2

Referee #2 (Comments on Novelty/Model System for Author):

De Kerckhove et al. identified Ago-2-bound miRNAs in mouse skin wounds, which included- among others - miR-223. This miRNA was upregulated during the early inflammatory phase of wound healing and is expressed mainly by neutrophils. Functional studies revealed that miR-223^{-/-} mice have impaired healing of sterile wounds, but enhanced healing of S. aureus infected wounds, most likely due to stronger activation of neutrophils and enhanced production of IL-6 by these cells. The potential therapeutic relevance of these results was demonstrated by knock-down of miR-223 at the wound site and by application of miR-223^{-/-} neutrophils, which resulted in enhanced healing of S. aureus infected wounds.

A role of miR-223 in inflammation and infection control had previously been demonstrated by others, and IL-6 had previously been identified as a miR-223 target. Therefore, these aspects are not completely novel. However, a role of miR-223 in wound healing has not been demonstrated. In particular, the role of this miRNA in healing of infected wounds is novel and interesting and of potential medical importance. However, there are also a few problems with the manuscript, which are summarized below.

Comment 1: 1.) *The authors should provide more information on the miR-223^{Y/-} mice in Materials and Methods. In particular, it should be mentioned that these hemizygous mice are completely deficient in miR-223 (at least according to the original publication).*

Response: As suggested, we have added information regarding the *miR-223* locus to the Materials and Methods (page 23, line 4 to 7).

Comment 2: 2.) *Since the manuscript focuses on miR-223, the authors should show expression of this miRNA during the whole time course of wound healing. There may be a second peak of miR-223 expression and this would be important for the interpretation of the wound healing data in the mutant mice.*

Response: We have now analyzed the expression of *miR-223* at wound sites in WT mice on days (d) 1, 3, 7, 10, and 14 after injury and in intact skin, and find that the expression of *miR-223* peaked at d 1 and was decreased by d 7 thereafter. The expression levels of *miR-223* at d 10 and 14 in wound sites of WT mice were very low, similar to that in intact skin (undetectable levels), because *miR-223* is expressed by inflammatory cells (i.e. neutrophils, macrophages) (Appendix Fig S2).

Comment 3: 3.) *It should be clarified that the early increase in miR-223 in skin wounds results from the infiltration of neutrophils and is most likely not a result of a real upregulation in immune cells.*

Response: It was reported that neutrophils mainly migrate to aseptic murine skin wound sites on d 1: the acute inflammation phase during skin wound healing, and therefore it is difficult to detect other immune cells (macrophages, lymphocytes) at this timepoint. Macrophages appeared in wound sites (Fig EV2) at d 3 and we found that *miR-223* was expressed in wound infiltrated macrophages at d 3 after injury (Fig 3D). Taken together, we suspect that the expression of *miR-223* in aseptic skin wound sites at the early timepoint (d 1 after injury) was predominantly from wound-infiltrated neutrophils, because macrophages had not migrated at d 1 after injury. On days 3 and 7, neutrophils and macrophages are present in wound sites, so that *miR-223* might be expressed by both cell types. We are currently investigating the function of *miR-223* in macrophages in skin wound healing.

Comment 4: 4.) *Fig. 2A: The authors should mention which antibody they used for the detection of neutrophils - Ly6G? In addition, they should mention in the legend that the area indicated with a reactange is shown at high magnification below.*

Response: As suggested by you and another reviewer, we have added information regarding the neutrophil antibody (Ly6-G and Ly6-C) and the rectangle in the Fig 3A and 3B (previously Fig 2A and 2B) legend (page 50, line 21 to 22) (page 51, line 4 to 5) and Appendix Table S5.

Comment 5: 5.) *Fig. 2D: The information in this figure is limited, since there is no comparison with other cells at the wound site, in particular fibroblasts and keratinocytes. Given the delayed reepithelialization in the mutant mice, it is particularly important to determine if this is a cell autonomous effect that results from expression of miR-223 in keratinocytes or a secondary effect resulting from enhanced inflammation (more likely).*

Response: We think that *miR-223* is only expressed by inflammatory cells and not by fibroblasts and keratinocytes, because *miR-223* was not expressed by *PU.1^{-/-}* mice that had no inflammatory responses at skin wound sites because they lack neutrophils, macrophages, and lymphocytes (Fig 2B) (page 4, line 15 to 17). We confirmed that wound-infiltrated neutrophils predominantly express *miR-223* at d 1 after injury and not keratinocytes by using *in situ* hybridization (ISH) (Fig 3A and 3B). With regard to the delayed re-epithelialization of *miR-223^{Y/-}* mice, we, like you, think that this is a consequence of increased acute inflammatory responses (secondary effect).

Comment 6: 6.) *Fig. 2D-H: The authors should confirm that miR-223 is indeed not expressed in the mutant mice. Wound healing is a combination of reepithelialization and wound contraction - is contraction also affected in the mutant mice?*

Response: We confirmed that the expression of *miR-223* was not expressed at wound sites in *miR-223^{Y/-}* mice at d 1, 3, and 7 after injury (undetectable expression level of *miR-223* in *miR-223^{Y/-}* mice) (see next page Fig 1 for reviewer only).

Figure 1 (reviewer only). Expression of *miR-223* in skin wound healing of WT and *miR-223^{Y/-}* mice

Expression of *miR-223* in murine skin wound healing measured by qPCR relative to 5S rRNA (n = 4 - 6). Data information. All values represent the mean ± SD.

As suggested by you and another reviewer, we investigated the expression of α -smooth muscle actin (α SMA), a marker of myofibroblast wound contraction, at skin wound sites. We found that the expression of α SMA was markedly decreased at d 7 in aseptic wound sites of *miR-223^{Y/-}* mice. We have modified the text accordingly (page 9, line 12 to 20) and Fig EV1D and EV1E.

Comment 7: 7.) Fig. 3B is not convincing - there is only a statistically significant difference at the 3h time point and at the 3d time point - this needs to be formulated more carefully. The different functionality of the neutrophils may be more important than this minor difference in number. I am also not convinced that there is impaired resolution, since no difference was seen at day 7.

Response: We agree with your comments and have revised the paragraph regarding the onset and resolution of inflammatory responses in WT and *miR-223^{Y/-}* mice according to your advice. In Fig 4A and 4B (previously Fig 3A and 3B), we wanted to show the neutrophil influx into the wound site over time using EGFP green fluorescent labeled neutrophils. Kim and colleagues (J Invest Dermatol, 2008), reported that neutrophil influx after skin wounding in *lys-EGFP* mice increased most rapidly over the initial 12 h and reached a maximum between d 1 and d 3. It then decreased precipitously at d 5 (Fig 2 and 3 in Kim et al., J Invest Dermatol, 2008). Our results are similar to theirs; at 12 h in WT mice, the neutrophil influx begins to increase and at d 3 the influx peaks. In contrast, the rate of influx in *miR-223^{Y/-}* mice appeared somewhat slower (3 h) although it also peaked at d 3. By d 3, the influx rate of *miR-223^{Y/-}* mice had surpassed that of WT mice, and the neutrophil influx became excessive. We also modified the representative results of *in vivo* fluorescent images of EGFP-expressing neutrophils in skin wound sites as reflected in the related graph (Fig 4A).

Comment 8: 8.) Fig. S3A: Please show representative stainings.

Response: We have added representative images of IHC for F4/80 (Fig EV2A, previously Fig S3A).

Comment 9: 9.) The authors should show *Il-6* mRNA levels in non-stimulated and activated neutrophils of wt and *miR-223* mutant mice - this would further support the regulation of *IL-6* by *miR-223*. The qPCR shown in Fig. 4B only shows that *Il-6* expression is enhanced in total wounds of *miR-223* mutant mice, which may be secondary to the enhanced numbers of neutrophils (and not a real regulation by the miRNA).

Response: We found no significant difference in numbers of neutrophils at d 1 in the wound sites of WT and *miR-223^{Y/-}* mice using *in vivo* imaging analysis (Fig 4A and 4B). Our ISH study demonstrated that *miR-223* was only expressed by neutrophils in wound sites at d 1 (Fig 3A and 3B). Taken together, we suspect the cause of increased *Il6* expression at 1 d in the wound sites of *miR-223^{Y/-}* mice might be associated with the regulation of *miR-223* in neutrophils.

Comment 10: 10.) The upregulation of *IL-6* is unlikely to explain the impaired healing in *miR-223* mutant mice (*IL-6* knockout mice have impaired wound healing; Lin et al., 2003; *IL-6* promotes wound healing in glucocorticoid-treated mice; Gallucci et al., 2001). Therefore, the mechanism

underlying the impaired healing in the miR-223 mutant mice under sterile conditions remains unclear. This should at least be discussed.

Response: As you suggest we have no more fully, discussed the function of IL-6 in skin wound healing and inflammatory responses in the text. We understand that IL-6 is required for skin wound healing based on earlier IL-6 KO study (Gallucci et al, FASEB J, 2000; Lin et al, J Leukoc Biol, 2003). However, it is also the case that excess IL-6 causes inflammatory diseases, leading to the use of IL-6 receptor antibody (Tocilizumab) as a therapeutic agent against Castleman disease and rheumatic diseases in the clinic (Yoshizaki et al, Hematol Oncol Clin North Am, 2018, PMID: 29157617) (Rubbert-Roth et al, Rheumatol Ther, 2018, PMID: 29502236). Therefore, it is important to control the amount of IL-6 at inflamed sites. We have modified the text accordingly (page 20, line 20 to page 21, line 5).

Comment 11: 11.) Fig. 5A and G: In addition to the macroscopic analysis, the authors should show H/E-stained sections from 7-day and 14-day wounds (and ideally use them to determine if there is an effect on reepithelialization and contraction). Given the rather high error bars, analysis of these histological parameters would clearly strengthen the data. At least one would like to get an idea about the histological features of the healing and healed wounds.

Response: Thank you for this suggestion. We have performed histological analysis on *S. aureus*-infected, neutrophil-transplanted, and *miR-223* AS ODN-treated skin wound sites. Re-epithelialization in *S. aureus*-infected wound sites of *miR-223*^{wt} mice showed enhanced re-epithelialization at d 3 and 7 (Fig EV3C).

We found that total wound area and pathological post-infectious necrotic lesion at d 7 and area of scar sites at d 14 in *S. aureus*-infected, neutrophil-transplanted, and *miR-223* AS ODN-treated skin wound sites were significantly decreased, accompany by changing α SMA expression in granulation tissues (Fig 7H-7J) (Fig EV3) (Appendix Fig S4C-S4E). We see no α SMA expression cells in pathological postinfectious necrotic lesion. We have modified the text accordingly (page 13, line 15 to 20) (page 14, line 16 to 20) (page 16, line 10 to 18).

Comment 12: 12.) Fig. 5G,H: Is it possible to determine how long the neutrophils used for treatment remain in the wound tissue? The cells could be labeled for this purpose.

Response: We have observed how long transplanted neutrophils remain in the *S. aureus*-infected skin wound sites using WT EGFP-expressing neutrophils purified from the bone marrow of lys-EGFP mice using magnetic beads (see the Materials and Methods). Large numbers of EGFP-expressing neutrophils were retained in *S. aureus*-infected skin wound sites at d 1 (Appendix Fig S4A). However, there were very low numbers of EGFP-expressing neutrophils in *S. aureus*-infected skin wound sites remaining at d 3, and none at d 7. These results indicate that transplanted neutrophils can remain in skin wound sites up to 3 d after transplantation.

Comment 13: 13.) Fig. 6: The authors should verify that *miR-223* is indeed downregulated by the ODNs at the wound site and check if IL-6 is upregulated.

Response: We verified the effect of *miR-223* AS ODN using the *S. aureus*-infected skin wound healing model. The expression of *miR-223* at *miR-223* AS ODN-treated skin wound sites was significantly reduced compared with controls at 6 h and 1 d after injury (Fig 7D).

We also investigated the expression of *Il6* using qPCR and found it to be significantly increased at d 1 in *miR-223* AS ODN-treated *S. aureus*-infected skin wound sites compared with controls (Fig 7E). We have modified the text accordingly (page 15, line 18 to page 16 line 7) and Fig 7C to 7E.

Comment 14: 14.) The paragraph describing the results shown in Fig.7 includes various errors in spelling and grammar and is therefore difficult to read. Most importantly, the results do not allow the conclusion that *miR-223* is regulated by C/EBP α in response to *S. aureus*. To text this possibility, the authors would have to overexpress C/EBP α and determine if the PGN-mediated decline in *miR-223* is rescued. Therefore, the scheme shown in Fig. 6G is not fully supported by the data.

Response: We have now obtained a full-length human C/EBP α overexpression vector (Clone ID: OHu20497C, GenScript, Piscataway, NJ) and transfected it into differentiated HL-60 (dHL-60) cells but, unfortunately, we could not establish C/EBP α overexpression dHL-60 cells (see below Fig 2 for reviewer only).

Figure 2 (reviewer only). Expression of *CEBPA* and miR-223 measured by qPCR. Expression of *CEBPA* (A) and *miR-223* (B) in control vector (pcDNA3.1)-transfected and full length human *C/EBPα* over expression (OE) vector-transfected dHL-60 after stimulation with PGN for 6 h measured by qPCR relative to B2M (n = 6). All values represent the mean ± SD.

Generally, it is difficult to transfect mature immune cells with plasmids (pcDNA3.1; 5428 bp, *C/EBPα* overexpression vector; 6523 bp) but not ODN (18-mer) because of the low efficiency. And we were also concerned that overexpression of *C/EBPα* in dHL60 cells might have detrimental effects on mature neutrophil nature because *C/EBPα* regulates myeloid differentiation. Indeed, *c/ebpα*^{-/-} mice exhibited a lack of mature neutrophils (Zhang DE et al, Proc Natl Acad Sci U S A, 1997, PMID: 9012825). Overexpression of *C/EBPα* in non-differentiated HL60 cells triggered them to become mature granulocytes (Radomska HS et al, Mol Cell Biol, 1998, PMID: 9632814). The expression level of *C/EBPα* was constant in the bone marrow-derived cells and decreased in mature neutrophils (Bjerregaard MD et al, Blood, 2003, PMID: 12560239).

As an alternative strategy and to verify Fig 8H (previously Fig 6G), we performed a ChIP assay to investigate *C/EBPα* binding to the *miR-223* promoter site. The binding activity of *C/EBPα* was significantly decreased at 6 h after PGN stimulation compared with non-stimulated dHL-60 cells (Fig 8F and 8G) (Fig EV5B and EV5C). Unfortunately, we could not verify the full-length human *C/EBPα* overexpression vector; however, we conclude that the expression of *miR-223* in neutrophils might be controlled by *C/EBPα* after PGN stimulation. We have modified the text accordingly (page 18, line 7 to 11) and Fig 8F and 8G, Fig EV5B and EV5C.

Referee #2 (Remarks for Author):

This is an interesting manuscript and the data are generally convincing. However, additional experiments and some rewriting are required for publication in EMM.

2nd Editorial Decision

18 July 2018

Thank you for the submission of your revised manuscript to EMBO Molecular Medicine. We have now received the enclosed reports from the referees that were asked to re-assess it. As you will see the reviewers are now globally supportive and I am pleased to inform you that we will be able to accept your manuscript pending the following final amendments:

1) Please include the additional control requested by referee 1. Please also address referees' comments in writing.

Please submit your revised manuscript within two weeks.

I look forward to reading a new revised version of your manuscript as soon as possible.

***** Reviewer's comments *****

Referee #1 (Remarks for Author):

The authors have greatly improved the paper, in particular by including new results about the granulation tissue formation and epithelialization processes. However these results should be summarized and discussed in the Discussion.

Likewise the results related to PGN and IL-6 should be better discussed. Again the relevance of using both PGN and miR-223 AS ODN is lacking.

The authors have used a rabbit polyclonal α -smooth muscle actin antibody from ABCAM. In the datasheet of the provider this antibody recognizes at least one additional unspecific band at 75 kDa by western blotting. The results should be confirmed by using the worldwide recognized mouse monoclonal α -smooth muscle actin antibody (clone 1A4) that should be used as a biotinylated antibody (direct immunostaining to overcome the species issue).

The fact that α -smooth muscle actin is a marker of contractile myofibroblasts has not been first demonstrated by this group of research. Therefore appropriate references should be quoted (e.g Gabbiani's publications).

Referee #2 (Remarks for Author):

The authors have performed new experiments to address my comments and they have significantly revised the manuscript. These changes have further improved the quality of the manuscript. This is a very important study and the work is of high technical quality.

2nd Revision - authors' response

28 July 2018

Reviewer comments:

Referee #1 (Remarks for Author):

Comment 1: *The authors have greatly improved the paper, in particular by including new results about the granulation tissue formation and epithelialization processes. However these results should be summarized and discussed in the Discussion.*

Response: Thank you for your interesting comments. With regard to the alteration of re-epithelialization and granulation tissue formation in each model, we think that this is a consequence of increased acute inflammatory responses (secondary effect), because *miR-223* could be not expressed in wound-infiltrated fibroblasts and keratinocytes. We have modified the text accordingly (p21, lines 19 to p22, line 4)

Comment 2: *Likewise the results related to PGN and IL-6 should be better discussed.*

Response: It was reported that murine neutrophils produced IL-6 after PGN stimulation (Strassheim D et al, J Immunol, 2005, PMID: 15944314). We have modified the text accordingly (p22, lines 7 to 8).

Comment 3: *Again the relevance of using both PGN and miR-223 AS ODN is lacking.*

Response: Even if *miR-223* expression is strongly suppressed by *miR-223* AS ODN, when PGN stimulation (*S. aureus* recognition) is received, positive feedback occurs; thus, through increased IL-6 production, infection control becomes more effective. We have modified the text accordingly (p17, lines 18 to 21).

Comment 4: *The authors have used a rabbit polyclonal α -smooth muscle actin antibody from ABCAM. In the datasheet of the provider this antibody recognizes at least one additional unspecific band at 75 kDa by western blotting. The results should be confirmed by using the worldwide recognized mouse monoclonal α -smooth muscle actin antibody (clone 1A4) that should be used as a biotinylated antibody (direct immunostaining to overcome the species issue).*

The fact that α -smooth muscle actin is a marker of contractile myofibroblasts has not been first demonstrated by this group of research. Therefore appropriate references should be quoted (e.g Gabbiani's publications).

Response: We observed an additional nonspecific band at 75 kDa in 3T3 cell lysates by western blotting (ab5694, Abcam). In contrast, no nonspecific band was observed for murine heart tissue homogenate, suggesting that the nonspecific band in the murine sample was only detected in the cancer cell line lysate; therefore, this antibody could be useful to detect α SMA in normal murine tissues. Additionally, this rabbit-derived α SMA antibody is utilized worldwide in immunohistochemistry for murine samples to an extent similar to mouse monoclonal α SMA antibody (clone 1A4). Recently, Plikus and colleagues showed that murine skin wound-infiltrated myofibroblasts were identified using the same antibody (Plikus et al, Science, 2017, PMID: 28059714, see Fig. 2A) (we confirmed antibody information via supplemental information in this paper and by direct confirmation from Drs. Plikus and Guerrero-Juarez). Moreover, we performed western blotting and confirmed a highly specific band that corresponded with α SMA protein (see below Fig. 1 for reviewer only). Therefore, we are confident that this anti- α SMA antibody (ab5694, Abcam) can be used for the immunohistochemistry of murine skin wound slices. We understand mouse monoclonal α SMA antibody (clone 1A4) is utilized for pathologic diagnosis of human tissue in soft tissue tumors such as leiomyoma and so on. Similarly, as our study is of murine wound tissues, we preferred to select rabbit-derived antibody rather than mouse-derived antibody.

Figure 1 (reviewer only). Expression of α SMA protein in day 7 murine skin wounds of WT mice.

We performed western blotting using polyclonal rabbit α SMA antibody (1:3000) (ab5694, Abcam) with overnight incubation at 4°C and a blocking time of 2 hours at room temperature (PVDF Blocking Reagent, TOYOBO). Then the secondary antibody (anti-Rabbit IgG HRP-linked whole antibody, GE Healthcare) (1:10,000) was incubated for 1 hour at room temperature. Protein bands were visualized by chemiluminescence (ImmunoStar LD, Fujifilm Wako Pure Chemical Corp.), and LAS3000 mini (exposure time: 1 second) (Fujifilm). The predicted band size of α SMA protein is 42 kDa. M, marker; lane 1, day 7 murine skin wound homogenate (10 μ g).

With regard to the reference for α SMA, we have replaced the reference (p9, line 16).

Referee #2 (Remarks for Author):

The authors have performed new experiments to address my comments and they have significantly revised the manuscript. These changes have further improved the quality of the manuscript. This is a very important study and the work is of high technical quality.

Corresponding Author Name: Ryoichi Mori

Manuscript Number: EMM-2018-09024